# Distinct prefrontal top-down circuits differentially modulate sensorimotor behavior

Rafiq Huda [1✉], Grayson O. Sipe [1], Vincent Breton-Provencher [1], K. Guadalupe Cruz [1], Gerald N. Pho [1], Elie Adam[1], Liadan M. Gunter[1], Austin Sullins [1], Ian R. Wickersham [2] & Mriganka Sur [1✉]

Sensorimotor behaviors require processing of behaviorally relevant sensory cues and the ability to select appropriate responses from a vast behavioral repertoire. Modulation by the prefrontal cortex (PFC) is thought to be key for both processes, but the precise role of specific circuits remains unclear. We examined the sensorimotor function of anatomically distinct outputs from a subdivision of the mouse PFC, the anterior cingulate cortex (ACC). Using a visually guided two-choice behavioral paradigm with multiple cue-response mappings, we dissociated the sensory and motor response components of sensorimotor control. Projection-specific two-photon calcium imaging and optogenetic manipulations show that ACC outputs to the superior colliculus, a key midbrain structure for response selection, principally coordinate specific motor responses. Importantly, ACC outputs exert control by reducing the innate response bias of the superior colliculus. In contrast, ACC outputs to the visual cortex facilitate sensory processing of visual cues. Our results ascribe motor and sensory roles to ACC projections to the superior colliculus and the visual cortex and demonstrate for the first time a circuit motif for PFC function wherein anatomically non-overlapping output pathways coordinate complementary but distinct aspects of visual sensorimotor behavior.

[1] Picower Institute for Learning and Memory, Department of Brain and Cognitive Sciences, Massachusetts Institute of Technology, Cambridge, MA 02139, USA. [2] McGovern Institute for Brain Research, Massachusetts Institute of Technology, Cambridge, MA 02139, USA. ✉email: rafiq.huda@rutgers.edu; msur@mit.edu

The behavioral repertoire of animals is highly enriched by their ability to learn how to respond to sensory cues to achieve goals such as reward[1–10]. Though seemingly simple, goal-oriented sensorimotor behaviors require coordination of multiple processes. Animals receive a deluge of environmental information at any given moment and can express a wide range of motor behaviors. Hence, sensorimotor control requires attentional mechanisms that prioritize processing of relevant sensory cues and select task-appropriate responses. Studies over the past decades have identified the prefrontal cortex (PFC) as a crucial nexus for coordinating sensorimotor behaviors. Specifically, the PFC is thought to generate control signals that facilitate task-specific processing[11–18]. However, a fundamental outstanding question is how the anatomical organization of inputs to and outputs from the PFC enables its proposed role in sensorimotor control.

Previous work using electrical stimulation demonstrates that the same PFC area can both enhance the representation of cortical visual signals, a neurophysiological hallmark of visual attention, as well as facilitate motor responses[14,19–21]. However, electrical stimulation is non-selective and hence these studies do not address whether specific PFC cell populations underlie sensory and motor functions. At the same time, other work suggests that the activity of functionally distinct populations of PFC neurons correlates with distinct components of sensorimotor behavior[16,22]. An intriguing possibility is that sensory and motor functions are subserved by distinct PFC subpopulations that target specific downstream structures.

Recent studies have identified a PFC area in mice, the anterior cingulate cortex (ACC), which is functionally and anatomically poised to exert control over visually guided behaviors. The ACC receives inputs from the visual cortex (VC)[23] and exhibits visual responses at single-neuron and network levels[24,25]. Studies employing causal manipulations using chemogenetics or optogenetics show that ACC activity is important for optimal performance on visually guided tasks[18,26–28]. The ACC provides outputs to the VC and motor-related layers of the superior colliculus (SC)[27–31], a crucial midbrain structure for response selection and other functions[9,32–40]. Importantly, these outputs originate from non-overlapping populations of ACC projection neurons[29], raising the possibility that these output pathways differentially modulate sensorimotor behavior. However, optogenetic activation of ACC outputs to both the VC or SC enhances the gain of stimulus-driven responses in the VC[27,30], suggesting a role for both pathways in sensory processing. While the function of ACC outputs to the VC in modulating cortical visual processing is consistent with previous work[14,16], a similar role for ACC outputs to the SC is surprising. The intermediate and deep layers of the SC are known to modulate specific motor functions and sensorimotor responses[9,32,36,41–44]. Furthermore, pharmacological inactivation of the SC produces a strong deficit in visual attention without perturbing the associated modulation of visual cortical activity[45]. Thus, the precise contribution of ACC projections to the VC and SC in mediating sensory processing and motor responses remains unclear.

Elucidating the function of ACC outputs to the VC and SC in sensorimotor control requires probing their contributions to the underlying visual processing and motor response components. Here, we establish a two-choice behavioral paradigm that allows us to distinguish these processes by testing mice on different cue–response contingencies. Comparing behavioral deficits induced by projection-specific inactivation of ACC output pathways across task contingencies shows that ACC projections to the SC modulate specific motor responses, while projections to the VC contribute to sensory processing. Remarkably, we find that ACC outputs to the SC facilitate motor responses by reducing the

innate response bias of the SC. By using two-photon calcium imaging of ACC inputs/outputs, virus-mediated anatomical tracing, and optogenetics, we delineate specific circuit mechanisms mediating this novel effect.

## Results

**The caudal ACC is anatomically positioned to contribute to visual sensorimotor behavior.** The ACC is a midline structure that spans a large extent across the caudo-rostral axis[46]. Although recent studies have implicated the mouse ACC in visual behaviors[18,26–28,30], this PFC region is also associated with other diverse behavioral functions[47–49]. We used rabies virus-mediated anatomical tracing and two-photon calcium imaging to determine if the ACC contains a subregion specialized for visual sensorimotor behaviors. We performed retrograde tracing using modified rabies viruses[50] to identify sources of visual inputs to the ACC. Rabies viruses encoding GFP and tdTomato were injected into caudal and rostral ACC, respectively (Fig. 1a). Although both compartments received inputs from medial higher VC, corresponding to functionally defined anteromedial and posteromedial areas[51], as well as the lateral higher VC, the caudal ACC also received inputs from the primary VC (Fig. 1b, c). Moreover, each ACC compartment received prominent inputs from its contralateral hemisphere (Fig. 1d). This anatomical organization suggests that: (a) the VC provides visual information to the ACC; (b) visual information is integrated in ACC activity; and (c) visual information is exchanged between the two ACC hemispheres via callosal projections. We tested these predictions using two-photon calcium imaging of VC and callosal axons in the caudal ACC. We unilaterally injected GCaMP6s in the VC or the ACC, and placed a chronic cranial window over the caudal ACC to image visually evoked activity of axons (Fig. 1e). While VC axons responded to stimuli presented in the contralateral visual field (relative to the site of recording), callosal axons responded preferentially to ipsilateral stimuli (Fig. 1f, g and Supplementary Fig. 1). Importantly, these callosal axon recordings establish that ACC neurons are visually responsive even in naïve mice, and relay visual information to the opposite hemisphere. Hence, VC and callosal inputs provide information about contra- and ipsilateral visual fields to the ACC, respectively.

ACC projection neurons are known to provide outputs to the SC[29,30]. We used anatomical tracing to determine if these projection neurons localize to the same subdivision that receives VC inputs. Injection of a rabies virus encoding tdTomato in the SC showed a high density of back-labeled neurons in the ACC (Fig. 1h). SC-projecting ACC neurons (ACC-SC) were located predominantly in the caudal subdivision (Fig. 1i). This labeling was observed ipsilateral to the site of injection, establishing the unilateral nature of this projection pathway. Together, our anatomical and functional studies show that the caudal ACC integrates visual inputs from the two hemispheres and provides outputs to the SC (Fig. 1j), which could allow this PFC area to contribute to visual sensorimotor control.

**A sensorimotor behavioral paradigm for studying sensory processing and motor responses.** We tested the role of the caudal ACC in visuomotor control by designing a behavioral paradigm for head-fixed mice inspired by previous work[6,10]. This paradigm assesses sensory processing by requiring mice to detect lateralized visual stimuli (Fig. 2a). Mice reported the spatial location of visual cues by rotating a trackball fixed along a single axis with their forepaws, additionally allowing us to study motor responses. To establish the relationship between ball rotations in head-fixed mice and spontaneous orienting or turning movements in freely moving mice, we tracked the forepaws using

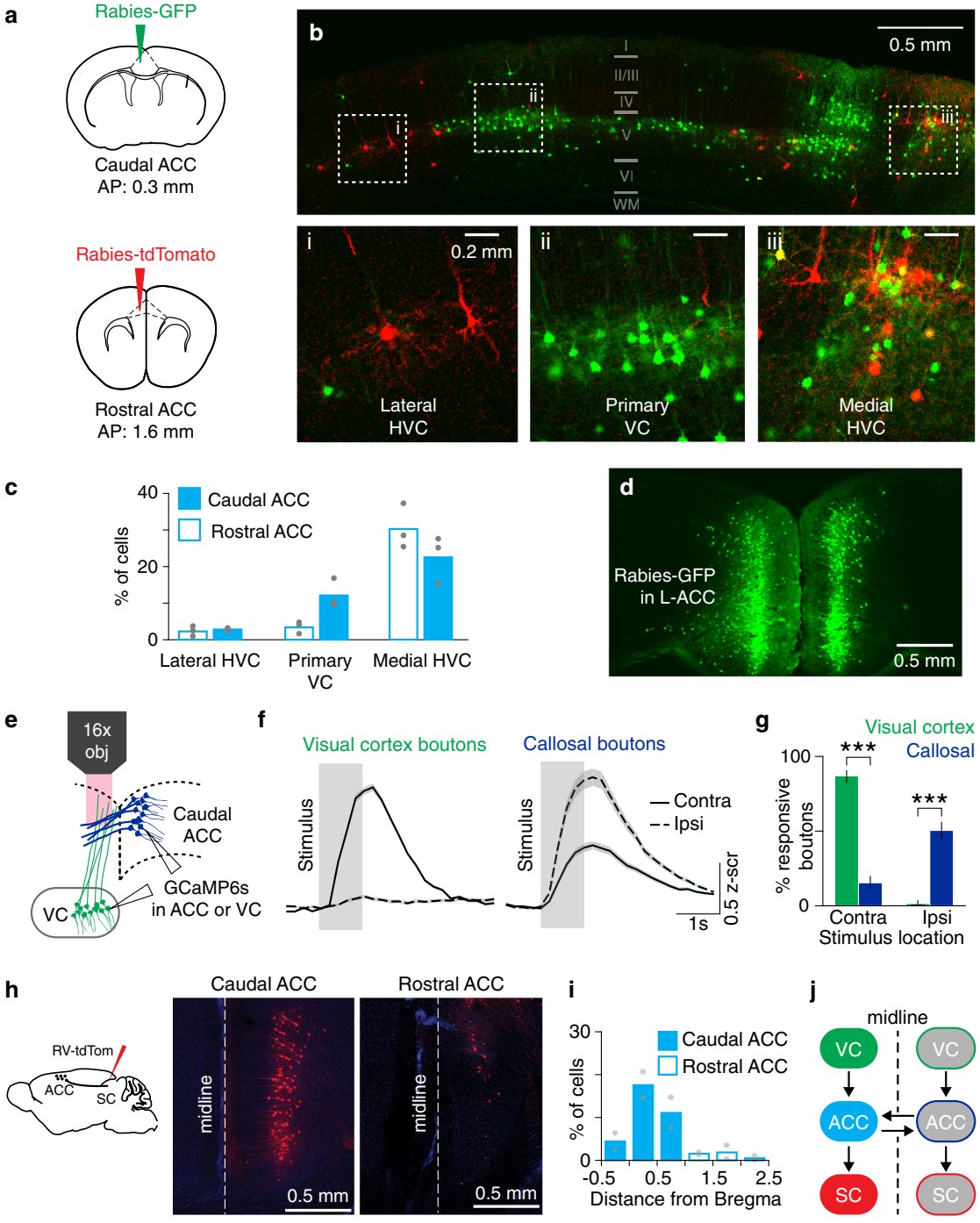

**Fig. 1 Anatomical and functional characterization of inputs to and outputs from the ACC. a** Rabies viruses encoding green fluorescent protein (GFP) or tdTomato were injected into caudal or rostral left anterior cingulate cortex (ACC), respectively. **b** Imaging of back-labeled neurons in the left visual cortex (green, GFP; red, tdTomato). Neurons in lateral higher VC (HVC), primary VC, or medial HVC are denoted by dotted squares (i-iii) on top and shown at higher magnification on the bottom. Similar results were observed across the three mice tested. **c** Proportion of back-labeled neurons in various VC subdivisions projecting to caudal (solid) or rostral (unfilled) ACC ($n = 3$ mice). **d** Back-labeled callosal neurons following injection in the left caudal ACC (L-ACC). Similar results were observed across the three mice tested. **e** Experimental setup for two-photon imaging via a 16× objective (obj) of GCaMP6s-expressing visual cortex (VC) or callosal axons in the ACC while head-fixed mice passively viewed visual stimuli (black square or grating, ~20°) presented in either hemifield. **f** Population-averaged responses of visually driven VC ($n = 268$ boutons from four mice) and callosal ($n = 309$ boutons from five mice) boutons to stimuli presented in contralateral (contra) or ipsilateral (ipsi) hemifields. Solid or dashed line is the mean response and shading shows the standard error of the mean. **g** Percent of visually driven VC and callosal boutons with preferential responses to contra (VC, $n = 233/268$ from four mice; callosal, $n = 48/309$ from five mice; $p \sim 10^{-72}$) and ipsi stimuli (VC, $n = 4/268$ from four mice; callosal, $n = 156/309$ from five mice; $p \sim 10^{-47}$). ***$p < 10^{-5}$ (Fisher's exact test, two sided). Errors bars are 95% binomial confidence intervals. **h** Rabies viruses encoding tdTomato were injected in the superior colliculus (SC), leading to labeling in the ACC (red, tdTomato; blue, DAPI). Similar results were observed across the two mice tested. **i** Proportion of back-labeled SC-projecting neurons located along the caudo-rostral axis of the ACC ($n = 2$ mice). **j** Schematic summary of anatomical tracing results.

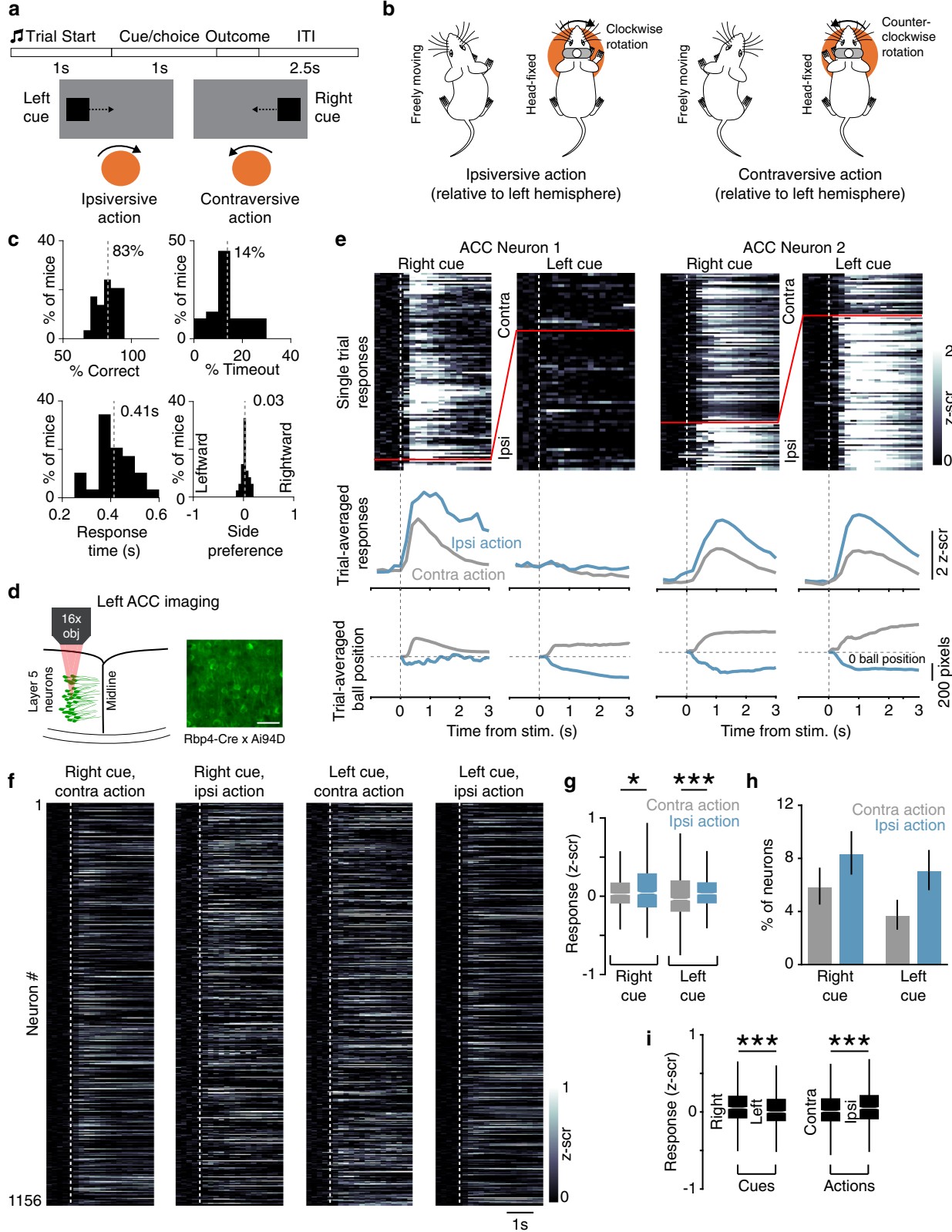

DeepLabCut[52] in either condition (Supplementary Fig. 2 and Fig. 2b). In freely moving mice, the paws were positioned opposite to the direction of the turn. During leftward turns, the paws were positioned to the right of the body on average; when the mice turned right, their forepaws were instead positioned to the left (Supplementary Fig. 2a, b). This opposing relationship was also seen in ball rotations of head-fixed mice. During

counterclockwise rotations of the ball, the forepaws were positioned to the left and clockwise rotations were associated with forepaw movements to the right (Supplementary Fig. 2c). Overall, this indicates that ball rotations in head-fixed mice are akin to orienting actions in freely moving mice (Fig. 2b).

We began these experiments with the "inward" cue–response contingency, in which mice rotated the ball to move the presented

**Fig. 2 Responses of ACC neurons in a visual sensorimotor task. a** Trial events during the task (top) and schematic showing the sensorimotor contingency during the inward task (bottom). Mice rotate the ball to center visual cues presented on either side of the screen. Correct responding moves the cue to the center of the screen, whereas incorrect responding moves it to the side. **b** Schematic showing the relationship between paw positions for spontaneous turns in freely moving mice and ball rotations in head-fixed mice. Actions are labeled contraversive and ipsiversive relative to the left hemisphere. **c** Behavioral performance of mice trained on this task for various optogenetic inactivation experiments ($n = 29$ mice). Dashed lines indicate mean. **d** Two-photon calcium imaging of GCaMP6s-expressing layer 5 neurons (green, GCaMP6s). Activity of 1156 neurons is reported from five expert mice (eight behavioral sessions, 1–3 from each mouse). Similar labeling was observed in all five mice tested. Scale bar, 50 μm. **e** Activity of two example ACC neurons during the task. Responses on right and left cue trials with contraversive (contra; gray) or ipsiversive (ipsi; blue) actions are shown. Responses on individual trials (rows in top color plots), trial-averaged responses on the indicated trial type (middle), and trial-averaged ball positions (bottom) are shown. Note that rows above and below the redline in color plots show trials with contra and ipsi actions, respectively. Vertical dotted lines correspond to the time of stimulus (stim) onset and horizontal dotted lines show a ball position of 0. **f** Session-averaged activity of individual ACC neurons for the indicated trial type is shown. Each row corresponds to the same neuron across the four plots. Dotted lines show stimulus onset. **g** Average z-scored (z-scr) responses on right and left cue trials with contraversive and ipsiversive actions (right cue, contraversive vs. ipsiversive: $p = 0.010$, $z = -2.56$; left cue, contraversive vs. ipsiversive: $p = 1.56 \times 10^{-7}$, $z = -5.25$; box plot elements: center line, median; box limits, upper and lower quartiles; whiskers, $1.5 \times$ interquartile range; outliers not shown; $n = 1156$ neurons from 5 mice). $*p < 0.05$, $***p < 0.005$, two-tailed Wilcoxon signed-rank test. **h** Percentage of ACC neurons with a significant difference in response to contraversive vs. ipsiversive actions for right and left cues ($n = 1156$ neurons from five mice). Error bars show 95% binomial confidence intervals. **i** Average responses on right vs. left cue trials ($p = 9.7 \times 10^{-8}$, $z = 5.33$) and contraversive vs. ipsiversive trials ($p = 5.9 \times 10^{-6}$, $z = -4.53$; $n = 1156$ neurons from five mice). $***p < 0.005$, two-tailed Wilcoxon signed-rank test. Box plot conventions are the same as **g**.

stimulus to the center of the screen (Fig. 2a). We defined actions as contraversive (contra) or ipsiversive (ipsi) based on the hemisphere of the brain under study, which is assumed to be the left side for all figures and text (see Table 1 for targeting details and behavioral performance of mice in specific experiments). Mice selected right cues with contraversive actions and left cues with ipsiversive actions (Fig. 2a, b). In later experiments, we trained mice on an "outward" contingency, in which mice moved cues to the outside of the screen. We used a reaction time task design that allowed mice to make a response as soon as they could after stimulus onset. This minimized potential confounds of short-term memory and extensive movement planning associated with delay tasks[53]. Experienced mice performed well on this task, with few timeouts (i.e., incomplete trials in which the ball is not moved to response threshold; note that timeout trials are excluded for calculating accuracy and are quantified separately) and responses with short latencies after stimulus onset (Fig. 2c). Moreover, mice did not exhibit significant side preferences and selected both actions equally (Fig. 2c).

**Activity of ACC-SC neurons during task performance.** We used two-photon calcium imaging to determine how ACC activity relates to sensorimotor control during this task. We placed a chronic window over the left ACC of Rbp4-Cre × Ai94D mice, which express the genetically encoded calcium sensor GCaMP6s in layer 5 excitatory neurons (Fig. 2d). We analyzed responses of single neurons during the four possible combinations of task cues and actions. Individual neurons responded to multiple task variables, yet response amplitudes were modulated for specific cues and actions. For example, the first ACC neuron shown in Fig. 2e selectively responded on right cue trials but showed a higher response for ipsiversive vs. contraversive actions. Neuron 2 in Fig. 2e was active on both right and left cue trials but had a higher response when an ipsiversive action was selected. This pattern of activation also held at the population level; left ACC activity was higher on ipsiversive than contraversive trials for both right and left cues (Fig. 2f–h).

To determine whether the activity of ACC neurons is modulated by the kinematic properties of movements in addition to the direction of action itself, we computed the velocity of ball rotations from the time of movement start until the ball position reached the response threshold. We did a median split of trials based on the ball velocity and compared task responses on trials with high and low velocity movements. Overall, ACC activity was similar for high and low velocity trials (Supplementary Fig. 3a, b).

This suggests that the observed modulation of activity on ipsiversive vs. contraversive trials is predominantly due to the action selected by the animal.

We determined how visual cues are represented by ACC neurons by comparing responses to right and left cues without regard to the action selected by the animal. Overall, ACC neurons had higher activity on right than left cue trials (Fig. 2i). We similarly compared responses on ipsiversive and contraversive action trials regardless of which visual cue was presented. This showed that ACC neurons respond preferentially to ipsiversive actions (Fig. 2i). Together, these analyses suggest that, as a population, left ACC neurons are preferentially activated by right cues and ipsiversive actions during the task.

We performed pathway-specific imaging to determine whether and how ACC-SC neurons, identified by injecting a synthetic retrograde tracer into the SC (Fig. 3a), respond to specific actions. Examination of single-neuron and population responses showed higher activation of ACC-SC neurons on ipsiversive trials (Fig. 3b–e). Like the overall ACC population, we found that the activity of ACC-SC neurons was not modulated by the velocity of ball movements (Supplementary Fig. 3c, d). Next, we used a decoding approach to probe whether ACC-SC activity predicted selected actions. We trained linear SVM (support vector machine) classifiers to distinguish between the two actions based on the single-trial activity of ACC-SC neurons recorded across all sessions. Classifiers trained with the activity of ACC-SC neurons performed better than chance at predicting actions (Fig. 3f). To determine whether ACC-SC neurons contain more action information than the general ACC population, we trained separate SVM classifiers with the activity of ACC-SC and matching numbers of unlabeled neurons recorded during individual sessions. ACC-SC neurons had a higher action decoding accuracy than unlabeled neurons in 7/8 sessions (Fig. 3g). Together, these analyses suggest that ACC neurons convey information about ipsiversive actions to the SC.

**The SC and the ACC-SC pathway facilitate opposing actions.** How does the ipsiversive action information observed in ACC-SC neurons contribute to task performance? We addressed this question by first examining the role of activity in the SC itself. We virally expressed the inhibitory opsin Jaws[54] in the intermediate and deep layers of the left SC and delivered yellow light (593 nm) through an implanted optic fiber on a randomly selected subset of trials (Fig. 4a). Illuminating the brain with light in the absence of an opsin did not significantly change responses to either right or

**Table 1 Performance metrics of mice included in this study.**

| Mouse # | Experiment | Figure[a] | Implant side | Control accuracy (%)[b] | Non-laser accuracy (%) | Timeout (%) | Response time (s) |
|---|---|---|---|---|---|---|---|
| 61 | Light Ctrl | F-S4 | Left | 97 | 95 | 8 | 0.35 |
| 62 | Light Ctrl | F-S4 | Left | 87 | 93 | 16 | 0.50 |
| 72 | Light Ctrl | F-S4 | Left | 93 | 87 | 28 | 0.39 |
| 100 | Light Ctrl | F-S4 | Left | 98 | 95 | 12 | 0.32 |
| 120[c] | Light Ctrl | F-S4 | Right | 78 | 78 | 32 | 0.53 |
| 122 | Light Ctrl | F-S4 | Left | 93 | 94 | 10 | 0.57 |
| 15 | SC inh. | F-4 | Left | 65 | 61 | 4 | 0.44 |
| 68 | SC inh. | F-4 | Left | 76 | 76 | 12 | 0.46 |
| 120[c] | SC inh. | F-4 | Left | 76 | 62 | 20 | 0.48 |
| 125 | SC inh. | F-4 | Left | 77 | 67 | 14 | 0.36 |
| 126 | SC inh. | F-4 | Left | 82 | 78 | 20 | 0.41 |
| 21[c] | ACC-SC inh. | F-4 | Left | 85 | 69 | 6 | 0.41 |
| 23[c] | ACC-SC inh. | F-4 | Left | 92 | 77 | 5 | 0.35 |
| 103 | ACC-SC inh. | F-4 | Left | 94 | 80 | 8 | 0.28 |
| 104 | ACC-SC inh. | F-4 | Left | 98 | 93 | 14 | 0.36 |
| 110 | ACC-SC inh. | F-4 | Left | 85 | 73 | 5 | 0.26 |
| 116[c] | ACC-SC inh. | F-4 | Right | 75 | 65 | 3 | 0.35 |
| 38 | Imaging | F-2,3 | Left | 83 | NA | 16 | 0.49 |
| 39 | Imaging | F-2,3 | Left | 88 | NA | 19 | 0.35 |
| 41 | Imaging | F-2,3 | Left | 84 | NA | 24 | 0.30 |
| 42 | Imaging | F-2,3 | Left | 92 | NA | 20 | 0.37 |
| 43 | Imaging | F-2,3 | Left | 76 | NA | 13 | 0.44 |
| 8 | VC inh. | F-S7 | Left | 97 | 85 | 12 | 0.35 |
| 9 | VC inh. | F-S7 | Left | 95 | 85 | 18 | 0.34 |
| 20 | VC inh. | F-S7 | Left | 89 | 87 | 9 | 0.40 |
| 19 | VC inh. | F-S7 | Left | 97 | 85 | 17 | 0.45 |
| 33 | VC inh. | F-S7 | Left | 97 | 73 | 24 | 0.41 |
| 21[c] | ACC-VC inh. | F-5 | Right | 90 | 79 | 6 | 0.37 |
| 22 | ACC-VC inh. | F-5 | Right | 90 | 91 | 12 | 0.48 |
| 23[c] | ACC-VC inh. | F-5 | Right | 92 | 89 | 6 | 0.36 |
| 107 | ACC-VC inh. | F-5 | Left | 96 | 90 | 14 | 0.29 |
| 114 | ACC-VC inh. | F-5 | Right | 75 | 75 | 15 | 0.47 |
| 116[c] | ACC-VC inh. | F-5 | Right | 75 | 76 | 8 | 0.38 |
| 117[c] | ACC-SC Rev. | F-6 | Left | 81 | 71 | 6 | 0.35 |
| 118[c] | ACC-SC Rev. | F-6 | Left | 88 | 89 | 28 | 0.37 |
| 119[c] | ACC-SC Rev. | F-6 | Left | 71 | 69 | 12 | 0.42 |
| 127[c] | ACC-SC Rev. | F-6 | Right | 79 | 74 | 11 | 0.38 |
| 129[c] | ACC-SC Rev. | F-6 | Right | 76 | 68 | 18 | 0.38 |
| 153[c] | ACC-SC Rev. | F-6 | Right | 86 | 84 | 18 | 0.38 |
| 117[c] | ACC-VC Rev. | F-6 | Right | 81 | 80 | 22 | 0.38 |
| 118[c] | ACC-VC Rev. | F-6 | Right | 88 | 91 | 16 | 0.38 |
| 119[c] | ACC-VC Rev. | F-6 | Right | 71 | 72 | 6 | 0.36 |
| 127[c] | ACC-VC Rev. | F-6 | Left | 79 | 72 | 4 | 0.36 |
| 129[c] | ACC-VC Rev. | F-6 | Left | 76 | 71 | 16 | 0.41 |
| 153[c] | ACC-VC Rev. | F-6 | Left | 86 | 83 | 18 | 0.35 |

[a] First figure referencing data from each mouse.
[b] Accuracy on control trials from non-optogenetic sessions.
[c] Mice contributed data to two experiments.

left cues (Supplementary Fig. 4). There was no difference in the percentage of incorrect or timeout trials (i.e., trials in which mice fail to give a complete response; timeout trials were excluded when quantifying percentage of incorrect trials) or in the response time (Supplementary Fig. 4b). To assess the effect of SC photoinhibition, we compared the percentage of trials in which mice responded incorrectly on right cue/contraversive action and left cue/ipsiversive action trials under "laser" and "no laser" conditions. Photoinhibition of the left SC biased responses; inhibition increased incorrect responses on right cue trials but decreased incorrect responses on left cue trials (Fig. 4b). In other words, SC inactivation decreased contraversive actions and increased ipsiversive actions. These results are consistent with a recent neurophysiology study demonstrating that the activity of SC neurons encodes contraversive actions in a similar head-fixed task[55]. Reducing SC activity also decreased the timeout rate on

ipsiversive trials and increased the response time on contraversive trials (Fig. 4b). These results suggest that the SC facilitates contraversive actions during the task.

To determine if the SC facilitates ball responses regardless of specific learned task contingencies, we tested a separate cohort of untrained mice as they spontaneously moved the ball. Inactivation of the SC reduced contraversive and increased ipsiversive actions, which were similar to movements during the task (Supplementary Fig. 5). The fact that SC inactivation modulated actions during spontaneous movement shows that this structure exerts an innate response bias and facilitates specific ball movement directions in head-fixed mice. These results, combined with the effects of SC inactivation during the visually guided task (Fig. 4b), demonstrate that the SC facilitates contraversive responses.

How do direct ACC inputs modulate the contribution of the SC to action selection in this task? We addressed this question by

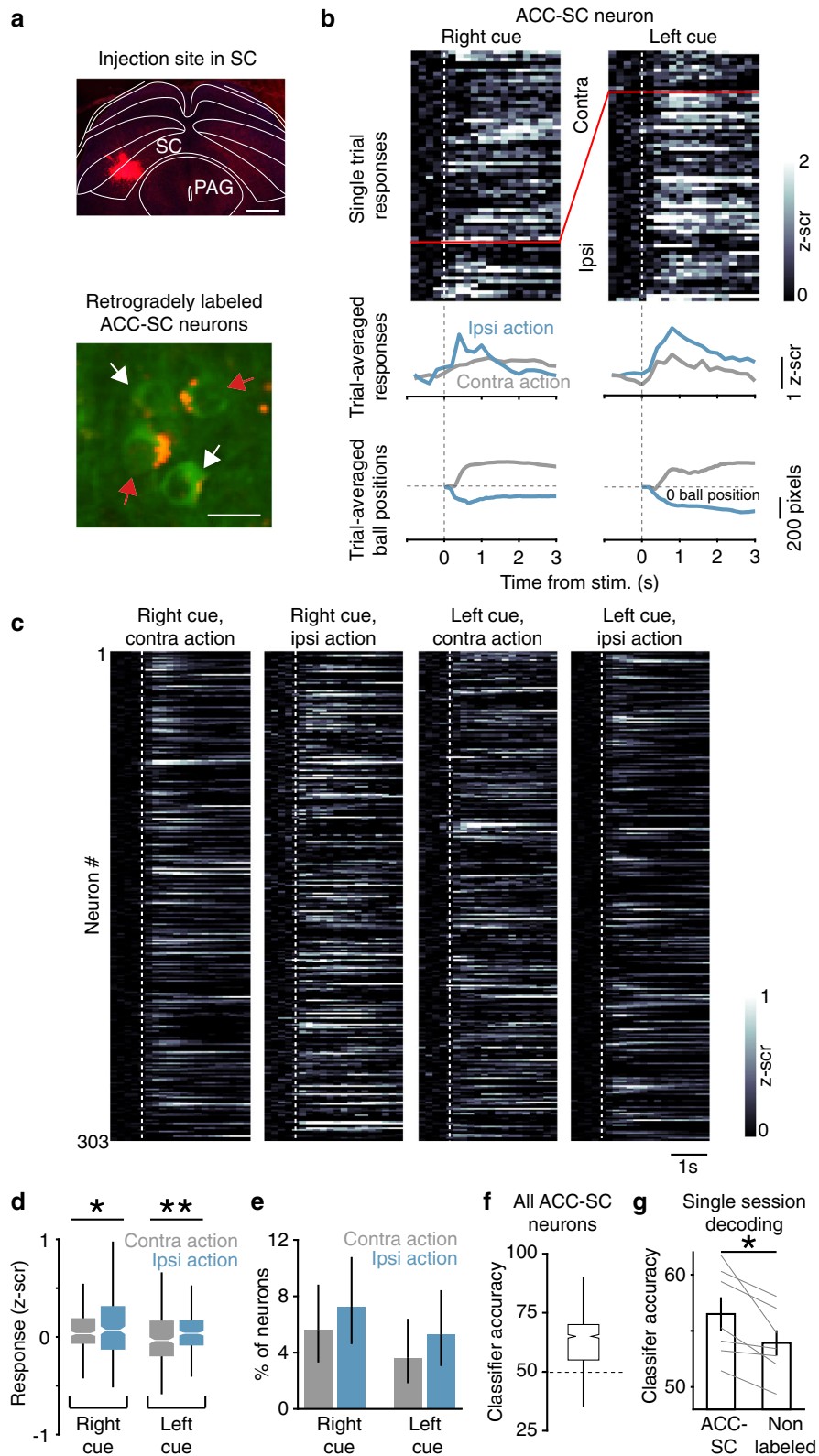

virally expressing Jaws in the ACC and implanting an optic fiber cannula over the left SC to target ACC-SC inputs for inactivation (Fig. 4c). ACC-SC inactivation increased incorrect responses on left cue trials and there was a trend for decreased incorrect responses on right cue trials (Fig. 4d). That is, inactivation of left ACC-SC inputs decreased ipsiversive and increased contraversive actions, opposite to that observed with SC inactivation (Fig. 4b).

Hence, ACC-SC inputs normally facilitate ipsiversive responses, consistent with task responses of ACC-SC neurons (Fig. 3d–g). Given that the left SC promotes contraversive responses in both trained and untrained mice (Fig. 4b and Supplementary Fig. 5), and that each hemisphere of the ACC targets the SC on the same side (Fig. 1h), these results suggest that the SC and the ACC-SC pathway facilitate opposite actions. Importantly, these findings

**Fig. 3 Task activity of ACC-SC neurons. a** ACC-SC neurons were identified by injecting red retrobeads in the superior colliculus (SC; green, GCaMP6s; red, retrobeads). Recordings were made from 303 ACC-SC neurons in five expert mice (eight behavioral sessions, 1–3 sessions per animal). White and red arrows show unlabeled and labeled neurons, respectively. Scale bars: top, 0.5 mm; bottom, 20 μm. Similar labeling was observed in the five mice tested. **b** Task responses of an example ACC-SC neuron. Color plot on top shows responses on right and left cue trials with contraversive (contra) and ipsiversive (ipsi) actions on individual trials (rows). Trial-averaged neuronal responses (middle) and trial-averaged ball positions (bottom) for the indicated trial types are shown. Vertical dotted lines show stimulus (stim) onset and horizontal dotted lines show ball position of 0. **c** Session-averaged activity of individual ACC-SC neurons for the indicated trial type. Each row corresponds to the same neuron across the four plots. Dotted line shows stimulus onset. **d** Average $z$-scored ($z$-scr) responses on right and left cue trials with contraversive and ipsiversive actions (right cues, contraversive vs. ipsiversive: $p = 0.046$, $z = -2.00$; left cue, contraversive vs. ipsiversive: $p = 0.0076$, $z = -2.69$; $n = 303$ neurons from five mice). *$p < 0.05$, **$p < 0.005$, two-tailed Wilcoxon signed-rank test. Box plot conventions are same as Fig. 2g. **e** Percentage of ACC-SC neurons with a significant difference in response to contraversive vs. ipsiversive actions for right and left cues ($n = 303$ neurons from five mice). Error bars are 95% binomial confidence intervals. **f** Classifier accuracy for action decoding with ACC-SC neurons recorded across all sessions. Dotted line shows chance model performance obtained by shuffling trial labels on the test set. Mean classifier accuracy ± the standard deviation over 1000 iterations of the model is shown ($n = 8$ sessions from 5 mice). **g** Classifier accuracy for action decoding on individual sessions with ACC-SC and unlabeled neurons ($n = 7$ sessions from four mice; $p = 0.018$, $z = 2.37$). *$p < 0.05$; two-tailed Wilcoxon signed-rank test. Error bars are mean ± standard error of the mean.

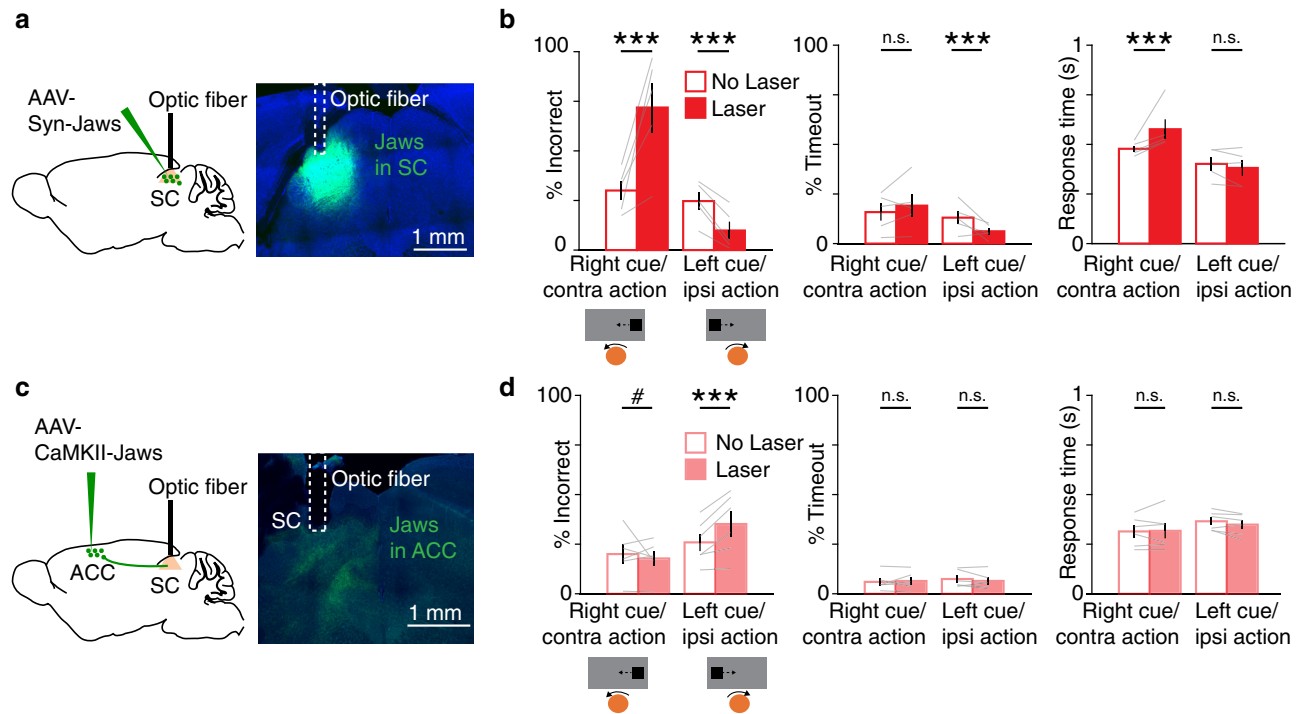

**Fig. 4 Effect of SC and ACC-SC inactivation on behavioral performance. a** AAV5-Syn-Jaws-GFP was injected in the SC and a fiber optic was implanted above the injection site (green, Jaws-GFP; blue, DAPI). Similar expression and fiber placement was observed across the 5 mice tested. **b** Behavioral performance for non-laser (unfilled) and laser (filled) conditions on right cue/contraversive action and left cue/ipsiversive action trials with SC inactivation ($n = 5$ mice). Incorrect performance (right cue, $p = 0.001$; left cue, $p = 0.001$), timeouts (i.e., trials with incomplete responses within the allotted time; right cue, $p = 0.077$; left cue, $p = 0.001$), and response time (right cue, $p = 0.001$; left cue, $p = 0.025$) are shown. ***$p < 0.005$; n.s., not significant. Significance evaluated at the Bonferroni-adjusted $p$ value of 0.025 with a two-tailed permutation test. **c** AAV5-CaMKII-Jaws-GFP was injected in the ACC and a fiber optic cannula was implanted in the SC (green, Jaws-GFP; blue, DAPI). Similar expression and fiber placement was found across the six mice tested. **d** Same as **b**, except for inactivation of ACC outputs to SC ($n = 6$ mice). Incorrect performance (right cue, $p = 0.061$; left cue, $p = 0.001$), timeouts (right cue, $p = 0.961$; left cue, $p = 0.561$), and response time (right cue, $p = 0.198$; left cue, $p = 0.092$) are shown. #$p = 0.061$. Significance testing same as **b**. Error bars in all panels are standard error of the mean.

also suggest that the ACC-SC pathway does so by modulating the innate response bias of the SC.

Since cortical projection neurons are predominantly glutamatergic, it is unclear how excitatory ACC-SC inputs implement a response bias opposite to the SC itself. We tested how the ACC influences activity in the SC. We photostimulated ChR2-expressing excitatory ACC neurons and used a 16-channel silicone probe to measure responses of SC neurons in the intermediate and deep layers of awake mice (Supplementary Fig. 6a). Photostimulation modulated the activity of 27% of SC units

(17/63 units from 3 mice) and led to a heterogeneous effect; while some SC units were excited, others were inhibited (Supplementary Fig. 6b, c).

These optogenetic manipulations could exert their effect indirectly by recruiting polysynaptic inhibitory pathways through the basal ganglia[56], excitatory-inhibitory commissural SC circuitry[57], or other mechanisms. Alternatively, these results may reflect direct effects on SC neurons. We used anterograde transsynaptic viral tracing[58] to label SC neurons targeted by the ACC. We injected tdTomato reporter mice with an AAV1 virus

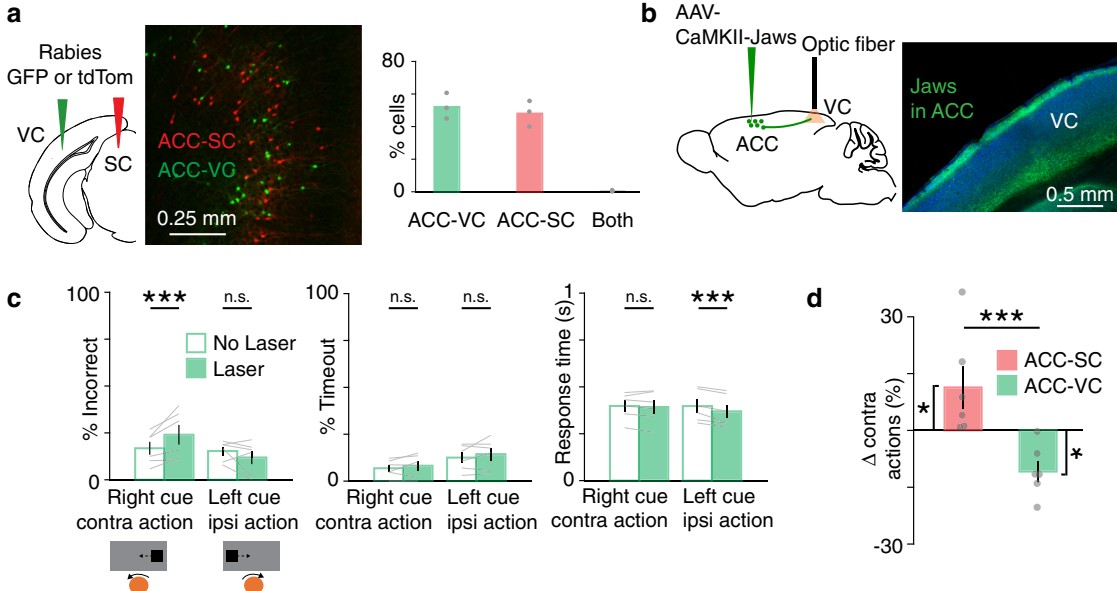

**Fig. 5 Effect of ACC-VC inactivation on behavioral performance. a** Dual-color retrograde tracing from the visual cortex (VC) and SC (green, GFP; red, tdTomato). Back-labeled ACC neurons projecting to the SC (red) or VC (green). Bar graph shows proportion of back-labeled neurons projecting to either or both areas ($n = 3$ mice). **b** AAV5-CaMKII-Jaws-GFP was injected in the ACC and axons in VC were targeted for inactivation (green, GFP; blue, DAPI). Similar expression was found across the six mice tested. **c** Behavioral performance for non-laser (unfilled) and laser (filled) conditions on right cue/contraversive action and left cue/ipsiversive action trials with inactivation of ACC outputs to the VC ($n = 6$ mice). Incorrect performance (right cue, $p = 0.002$; left cue, $p = 0.153$), timeouts (right cue, $p = 0.204$; left cue, $p = 0.312$) and response time (right cue, $p = 0.656$; left cue, $p = 0.006$) are shown. ***$p < 0.005$; n.s. not significant. Significance evaluated with two-tailed permutation test at the Bonferroni-adjusted $p$ value of 0.025. **d** Comparison of laser-induced change in contraversive actions with inactivation of ACC outputs to SC and VC (ACC-SC vs. zero, $n = 6$ mice, $p = 0.046$, $z = -1.992$; ACC-VC vs. zero, $n = 6$ mice, $p = 0.028$, $z = 2.201$; ACC-SC vs. ACC-VC, $p = 0.003$, $z = 2.802$). Comparison against zero with two-tailed Wilcoxon signed-rank test and comparison between ACC-SC and ACC-VC with one-tailed Wilcoxon rank-sum test. *$p < 0.05$; ***$p < 0.005$. Error bars are standard error of the mean in all panels.

expressing the Flpo recombinase in the ACC and performed immunohistochemistry against NeuN and GABA in slices containing the SC (Supplementary Fig. 6d, e). Overall, 6.6 ± 1.3% of SC neurons expressed tdTomato ($n = 5$ mice). tdTomato labeling was observed in both GABA- and non GABA-containing neurons, suggesting that ACC inputs target both excitatory and inhibitory SC neurons. Yet, we found an overrepresentation of GABAergic neurons in our sample of tdTomato positive neurons, as compared to the overall SC population (Supplementary Fig. 6f). Therefore, ACC neurons may recruit both excitatory and inhibitory neurons to modulate SC activity (Supplementary Fig. 6g), potentially in a task-dependent manner.

**ACC outputs to the VC facilitate performance on right cue trials**. Our results suggest that the ACC-SC pathway facilitates specific actions during the task. Sensory processing of behaviorally relevant stimuli is another key component of visual sensorimotor control. The ACC densely projects to the VC, modulates its activity, and facilitates performance on visually guided sensorimotor tasks[17,18,28,29]. Hence, we determined the role of the ACC-VC pathway in this task. First, we performed retrograde tracing to determine the organization of ACC outputs to the VC and the SC. Injection of red and green fluorophore expressing rabies viruses in the SC and VC, respectively, showed that non-overlapping populations of ACC projection neurons target either structure, suggesting functional differentiation in ACC outputs (Fig. 5a). To determine the contribution of the ACC-VC pathway, we first assessed the function of the VC in this task. Unilaterally inactivating the left VC by delivering yellow light onto Jaws-expressing neurons increased incorrect responses on right cue trials (Supplementary Fig. 7). While this

manipulation did not change the response time, there was an increase in the timeout rate on right cue trials, consistent with a role for the VC in detection of contralateral visual cues[59]. These results suggest that the ACC could facilitate performance on right cue trials by modulating activity in the VC. We directly tested this hypothesis by expressing Jaws in the ACC and photostimulating ACC output axons in the VC during the task (Fig. 5b). Inactivating the ACC–VC pathway increased incorrect responses on right cue trials (Fig. 5c). Directly comparing the effect of inactivating the ACC-SC and ACC-VC pathways showed that ACC outputs to SC increased and outputs to VC decreased contraversive actions associated with the right cue (Fig. 5d). Thus, ACC-SC and ACC-VC pathways oppositely control behavioral performance in this task and may differentially modulate sensorimotor behavior.

**ACC-SC and ACC-VC pathways differentially modulate sensory processing and motor responses**. How do ACC-SC and ACC-VC pathways control task performance? Optogenetic inactivation of these pathways can directly modulate the selection of specific actions or disrupt processing of visual stimuli that cue actions. We dissociated these possibilities by training a new cohort of mice on a reversed sensorimotor contingency, such that mice were required to move the presented visual cue outward to the side of the screen instead of the center (Fig. 6a). In this "outward" task, the action previously associated with a left cue was now cued by the right stimulus (and vice versa). Inactivation of the ACC-SC pathway caused a reversal in deficit and increased incorrect responses on right cue trials (Fig. 6b), as opposed to the deficit observed on left cue trials in the previous task contingency (Fig. 4d). In other words, ACC-SC inactivation decreased

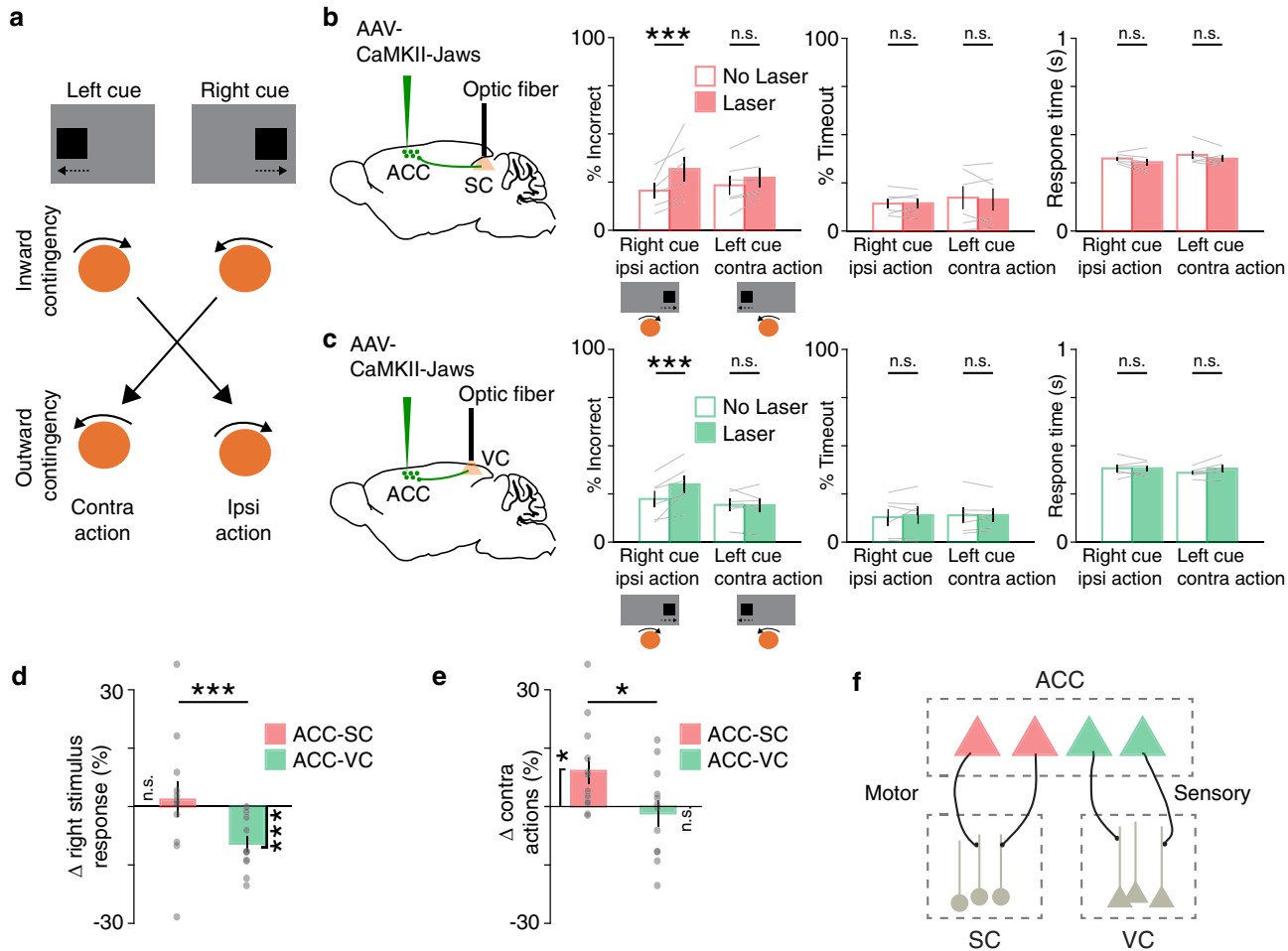

**Fig. 6 ACC outputs to SC and VC differentially modulate motor and sensory processing, respectively. a** Mice were trained on the outward sensorimotor contingency and were required to move left and right cues to the side of the screen. **b** Behavioral performance for non-laser (unfilled) and laser (filled) conditions on right cue/ipsiversive action and left cue/contraversive action trials with inactivation of ACC outputs to the SC in the outward task ($n = 6$ mice). Incorrect performance (right cue, $p = 0.001$; left cue, $p = 0.127$), timeouts (right cue, $p = 0.924$; left cue, $p = 0.927$), and response time (right cue, $p = 0.033$; left cue, $p = 0.138$) are shown. ***$p < 0.005$; n.s. not significant. Significance evaluated with two-tailed permutation test at the Bonferroni-adjusted $p$ value of 0.025. **c** Same as **b**, except for inactivation of ACC outputs to the VC ($n = 6$ mice). Incorrect performance (right cue, $p = 0.002$; left cue, $p = 0.786$), timeouts (right cue, $p = 0.499$; left cue, $p = 0.885$), and response time (right cue, $p = 0.869$; left cue, $p = 0.042$) are shown. Significance testing same as **b**. **d** Data from both tasks was combined. Effect of ACC-SC ($n = 12$ mice) and ACC-VC inactivation ($n = 12$ mice) on laser-induced change in right stimulus responses (ACC-SC vs. zero, $p = 0.638$, $z = -0.471$; ACC-VC vs. zero, $p = 0.002$, $z = 3.059$; ACC-SC vs. ACC-VC, $p = 0.004$, $z = 2.685$). ***$p < 0.005$; comparison against zero with two-tailed Wilcoxon signed-rank test, and comparison between ACC-SC and ACC-VC with one-tailed Wilcoxon rank-sum test. **e** Similar to **d**, except showing laser-induced change in contraversive responses (ACC-SC vs. zero, $p = 0.019$, $z = -2.353$; ACC-VC vs. zero, $p = 0.695$, $z = 0.392$; ACC-SC vs. ACC-VC, $p = 0.03$, $z = 1.876$). *$p < 0.05$; comparison against zero with two-tailed Wilcoxon signed-rank test, and comparison between ACC-SC and ACC-VC with one-tailed Wilcoxon rank-sum test. **f** Model summarizing the sensory and motor roles of ACC projections to the VC and SC, respectively. All error bars represent the standard error of the mean.

ipsiversive actions in both tasks. Thus, the ACC-SC pathway facilitates specific actions in either task contingency. In contrast, inactivating the ACC-VC pathway increased incorrect responses on right cue trials (Fig. 6c), as observed in the previous task (Fig. 5c). Hence, the left ACC-VC pathway is important for sensory processing of right stimuli in both tasks. We did not observe significant changes in timeouts or the reaction time with either manipulation in the outward task (Fig. 6b, c).

We further determined how deficits observed with ACC-SC and ACC-VC pathways map onto responses to specific visual cues or the direction of the action itself. We combined data across the two tasks and computed a right stimulus response index to compare how inactivation of ACC-SC and ACC-VC pathways affects responses associated with the right visual cue. Inactivation of the ACC-VC pathway significantly decreased responses cued by right stimuli, whereas inactivation of the ACC-SC pathway had no

consistent effect (Fig. 6d). Next, we computed a contra action index to determine how optogenetic inactivation modulates contraversive actions regardless of which stimulus cues them. While ACC-SC inactivation increased contraversive responses, we did not observe a significant change with ACC-VC inactivation (Fig. 6e). Moreover, we evaluated whether the various optogenetic manipulations employed modulate lower-level movement kinematics by analyzing the effect of optogenetic inactivation on ball trajectories. Movement trajectories were largely similar for control and laser trials (Supplementary Fig. 8). Thus, the ACC-SC pathway mediates selection of specific actions, whereas the ACC-VC pathway facilitates sensory processing in these tasks (Fig. 6f).

## Discussion

We demonstrate that the ACC modulates visual sensorimotor behaviors by using anatomically distinct but functionally

complementary populations of projection neurons to facilitate sensory processing (ACC-VC) and specific actions (ACC-SC). We show that VC inputs bring contralateral stimulus information and callosal inputs bring ipsilateral stimulus information to the caudal ACC (Fig. 1a–g and Supplementary Fig. 1). In turn, a subset of ACC neurons projects to the SC (Fig. 1h). Using a head-fixed, two-choice visuomotor task with responses akin to orienting actions (Fig. 2b and Supplementary Fig. 2), we show that the activity of ACC-SC neurons predicts ipsiversive actions (Fig. 3d–g). SC inactivation decreases contraversive responses during the inward task and spontaneous movements (Fig. 4a, b and Supplementary Fig. 5), consistent with a role for this area in response selection[9,37]. Surprisingly, inactivation of the ACC-SC pathway has the opposite effect and disrupts performance by decreasing ipsiversive responses (Fig. 4d). Importantly, by training mice on a reversed sensorimotor contingency (outward task), we demonstrate that ACC-SC inactivation consistently decreases ipsiversive responses regardless of the specific cue–response mapping (Fig. 6b, e). Hence, the ACC-SC pathway primarily contributes to motor responses. In contrast, ACC-VC inactivation decreases performance on right cue trials in either task (Figs. 5c and 6c), suggesting that it facilitates stimulus processing regardless of the associated response.

The SC is an integrative node within the broader midbrain selection network, serving as a key arbiter for response selection across multiple sensory and motor modalities in many species[9,32–36,42,60]. In our tasks, mice make responses by rotating a ball with their forepaws. We found key similarities between ball rotations in head-fixed mice and orienting movements in freely moving mice (Supplementary Fig. 2 and Fig. 2b), which likely reflect shared underlying neural substrates for both movements. Indeed, SC activity has a profound effect on the selection of ball rotation directions in both trained and untrained mice (Fig. 4b and Supplementary Fig. 5), as expected for orienting movements. Hence, our results suggest that ball rotation movements serve as a potential model system for studying orienting behavior in head-fixed mice.

Importantly, we find that unilateral SC inactivation does not increase inaction, but rather changes the likelihood of selecting specific responses (Fig. 4b). In other left–right response tasks, neurophysiological recordings show a subset of SC neurons respond maximally for contraversive responses and are often silent or inhibited for ipsiversive responses[9,55,61]. Similarly, causal activity manipulations produce response biases consistent with these activity measurements[41,42,61]. These results, combined with our own, suggest a "winner-take-all" interhemispheric competition model for selection in midbrain circuits, wherein the final response is determined by the SC hemisphere with the higher level of activity[9,62]. While the competition process underlying response selection likely involves an interplay between multiple systems, including intrinsic and commissural SC circuitry[60,63,64] and inputs from the basal ganglia[56], our results identify the important role of direct ACC projections to the SC in this process.

We found that SC and ACC-SC inactivation oppositely modulates behavioral performance in the inward task contingency (Fig. 4b, d). Several mechanisms could mediate this effect. In untrained mice, photostimulation of ACC neurons both excites and inhibits a subset of SC units (Supplementary Fig. 6a–c). Moreover, ACC outputs directly target inhibitory as well as excitatory SC neurons (Supplementary Fig. 6d–f). Hence, one possibility is that ACC projections recruit local inhibition in the left SC on left cue trials (Supplementary Fig. 6g), tipping the balance in favor of activity in the right SC and increasing the probability of selecting the left cue with an ipsiversive action. Alternatively, this effect may be mediated via commissural

excitatory SC neurons[63,64] that are targeted by ACC neurons. In this case, during a left cue trial, activation of left ACC-SC neurons would indirectly increase activity in the right SC (in addition to the cue directly activating the SC via inputs from the overlying sensory layer and other structures), thereby leading to a selection of the left cue. In both scenarios, ACC inputs contribute to response selection by modulating the relative levels of activity between the two SC hemispheres. A better understanding of the downstream anatomical targets of SC neurons that receive ACC inputs will help clarify these issues. Regardless of exact mechanisms, we clearly demonstrate that the ACC-SC pathway facilitates specific actions in this task, likely by modulating the innate response bias of its targeted SC hemisphere.

A subset of ACC-SC neurons was recently shown to provide collaterals to the lateral posterior (LP) thalamus and modulate cortical sensory processing in the VC[30]. This raises the possibility that ACC-SC neurons are anatomically and functionally heterogeneous. While we clearly demonstrate that direct ACC-SC outputs are critical for facilitating specific motor responses and contribute minimally to sensory processing, it is possible that the sensory function of this pathway was not engaged by the relatively simple visual stimuli used in this task. In contrast to the inward task, we found that the ACC-SC pathway is important for responses on right cue trials in the outward task contingency (Fig. 6b). Hence, task responses of ACC-SC neurons are likely shaped by specific sensorimotor contingencies.

Our results also support the hypothesis that the PFC exerts executive control over task responses by modulating activity in downstream motor structures. In the inward task, mice move their forelimbs in a direction that would orient them to the visual cue if they were freely moving (Supplementary Fig. 2 and Fig. 2b), similar to a "pro" response in eye movement or whole-body orienting tasks[43,65]. In the outward task contingency, mice make movements that orient them away from the visual cue, possibly similar to an "anti" response. The PFC is thought to play a crucial role in facilitating "anti" performance[65]. Our projection-specific inactivation of the ACC-SC pathway in the inward and outward task contingencies (Figs. 4d and 6b, e) are consistent with a role for the PFC in facilitating "anti" performance.

Although ACC outputs to the SC coordinate-specific responses, we found that the ACC uses an anatomically non-overlapping but functionally complementary population of projection neurons to facilitate sensory processing through the VC (Figs. 5c and 6c, d). Surprisingly, we found that areal VC inactivation only modestly affected behavioral performance, decreasing correct responding and increasing the timeout rate on right cue trials (Supplementary Fig. 7). Inactivation of the SC led to a more robust behavioral effect (Fig. 4b). A parsimonious interpretation of these findings is that focal viral inactivation of the VC, potentially targeting a subset of output pathways, reveals a relatively small contribution to performance in this task. At the same time, we found that the ACC-VC pathway facilitates responses to specific visual cues, demonstrating that ACC modulation of the VC is an important contributor to task performance. In future experiments, it will be important to use transgenic mice stably expressing opsins for large areal inactivation and projection-specific targeting of various VC output projections to test the full measure of VC contributions.

An important issue is whether ACC outputs to the VC and SC control behavioral performance by modulating sensory or motor components of the task. Previous studies have addressed this issue using delay tasks with stimulus presentation and motor response epochs separated by an intervening delay[2,5,66,67]. However, in such tasks, the cue–response mapping is known to the subject before stimulus presentation and movement planning can start with stimulus onset[68], making it difficult to

disambiguate whether temporally restricted inactivation affects behavior by disrupting sensory or motor processing. We instead addressed this issue by training a set of mice on the outward sensorimotor contingency and comparing the effect of projection-specific inactivation across task conditions. This showed that the ACC-VC pathway is important for responses associated with specific visual cues (Fig. 6c, f), suggesting it predominantly contributes to sensory processing. This is consistent with previous work demonstrating that the ACC-VC pathway modulates stimulus encoding by VC neurons in passively viewing mice and facilitates stimulus discrimination[17]. While the exact mechanisms mediating task performance through the VC are unclear, the ACC-VC projection may control VC outputs to multiple pathways that ultimately converge in the SC and other parts of the midbrain selection network[56,69,70].

Overall, our results highlight the importance of projection-specific manipulations in multiple task contingencies for dissecting circuit mechanisms underlying sensorimotor behaviors. By dissociating the contribution of individual outputs, our findings suggest a general organizing principle for PFC circuits wherein complementary behavioral functions are fulfilled by anatomically distinct output pathways, thereby enabling independent control over specific functions depending on task demands[71].

## Methods

**Animals**. All experimental procedures performed on mice were approved by the Massachusetts Institute of Technology Animal Care and Use Committee. Mice were housed on a 12 h light/dark cycle with temperature (70 ± 2 °F) and humidity (30–70%) control. Animals were group-housed before surgery and singly housed afterwards. Adult mice (>2 months) of either sex were used for these studies. In addition to wild-type mice (C57BL/6J), the following transgenic lines were used: Rbp4-Cre (MMMRC stock # 037128-UCD), Ai94(TITL-GCaMP6s)-D (Jackson Laboratory stock # 024104), and Ai65(RCFL-tdT)-D (Jackson Laboratory stock # 021875).

**Surgical procedures**. Surgeries were performed under isoflurane anesthesia (3–4% induction, 1–2.5% maintenance). Animals were given analgesia (slow release buprenex 0.1 mg/kg and Meloxicam 0.1 mg/kg) before surgery and their recovery was monitored daily for 72 h. Once anesthetized, animals were fixed in a stereotaxic frame. The scalp was sterilized with betadine and ethanol. For anatomical tracing experiments, we made a midline incision in the scalp using a scalpel blade. Depending on the experiment, rabies or AAV viruses were injected in the VC (AP: −3.5 mm, ML: 2.5 mm, DV: 0.5 mm), caudal ACC (AP: 0.3 mm, ML: 0.5 mm, DV: 0.5 mm), rostral ACC (AP: 1.6 mm, ML: 0.3 mm, DV: 1.3 mm), or the SC (at AP: −3.6 mm, ML: 1 mm, and DV: 1.5 mm) using a microinjector (Stoelting). After virus injection, the scalp was reclosed with sutures and skin adhesive (Vetbond).

The following surgical procedures were performed for optogenetics or imaging experiments. For experiments requiring the use of dental acrylic (fiber optic cannulae and chronic imaging windows), we removed a portion of the scalp using spring scissors, scraped away the periosteum membrane overlying the skull, and used a dental drill to abrade the skull to improve adhesion. For light control experiments, two mice expressed axonal GCaMP6s and four mice were wild type (Supplementary Fig. 4b). For photoinhibition of the SC during the inward task or spontaneous movements, we injected AAV5-hSyn.Jaws-KGC-GFP-ER2 (100 nL) in the intermediate/motor layer (AP: −3.6 mm, ML: 1 mm, DV: 1.5 mm) and implanted a fiber optic cannula (300 μm/0.39 NA core, CFM13L02, Thorlab) 0.2 mm dorsal to the injection site. Cannulae were secured on the skull using layers of Metabond and were protected with a dust cap until used for experiments. For modulating ACC outputs to the SC, we injected AAV5-CaMKII-Jaws-KGC-GFP-ER2 (University of North Carolina vector core) in the ACC and implanted a fiber optic cannula (300 μm/0.39 NA core or 400 μm/0.5 NA core, CFMXD02, Thorlabs) over the intermediate layer of the SC (AP: −3.6 mm, ML: 1 mm, DV: 1.3 mm). For optogenetic inactivation of the VC, we drilled a 3 mm craniotomy and made 8–12 injections (100 nL each) of an AAV5.CaMKII.Jaws-KGC-GFP-ER2 (University of North Carolina vector core) virus 0.5 mm below the surface in a grid pattern. Injections were centered on the left primary VC (centered at AP: −3.5 mm, ML: 2.5 mm; range AP: 4–3 mm, ML: 2.2–2.8 mm). For inactivation of ACC axons in VC, we injected AAV5-CaMKII-Jaws-KGC-GFP-ER2 in the ACC and implanted a 3 mm chronic window or a fiber optic cannula (400 μm/0.5 NA core) over the VC (centered at AP: −3.5 mm, ML: 2.5 mm, AP = 0, on pia). In a subset of experiments on the inward task, AAV5-CaMKII-Jaws-KGC-GFP-ER2 was injected bilaterally in the ACC and the VC or SC was targeted for axonal inactivation on either side of the brain. For projection-specific optogenetic inactivations during the

outward contingency task, we bilaterally injected AAV5-CaMKII-Jaws-KGC-GFP-ER2 in the ACC and implanted fiber optic cannulas over the SC and VC on either side of the brain (see Table 1 for details on which brain hemisphere was targeted for each experiment).

For all imaging experiments in the ACC, we drilled a 3 mm craniotomy over the midline (centered at AP: 0.5 mm and ML: 0 mm) and implanted a chronic imaging window assembled from two 3 mm coverslips glued to a 5 mm coverslip using a UV curable adhesive (Norland 61; double-chronic window). Two 3 mm coverslips were required to minimize movement artifacts for GCaMP recordings during task performance. To label SC-projecting ACC neurons (ACC-SC), we injected 100 nL of red retrobeads (Lumaflour) in the intermediate/motor layer of the SC (at AP: −3.6 mm, ML: 1 mm, DV: 1.7 mm). For axonal imaging experiments, 200 nL of AAV1-Syn-GCaMP6-WPRE-SV40 virus was injected unilaterally in the VC (AP: −3.5 mm, ML: 2.5 mm, DV: 0.5 mm) or the opposite ACC (AP: 0.5 mm, ML: 0.5 mm, DV: 0.5 mm).

For all experiments, after implantation of chronic windows or optic fiber cannulas, the skull was attached to a stainless-steel custom-designed headplate (eMachines.com) using Metabond. Animals were allowed to recover for at least 5 days before commencing water restriction for behavioral experiments.

**Virus-mediated anatomical tracing**. Standard histological techniques were used for analysis of retrograde and anterograde trans-synaptic tracing experiments and for post hoc verification of implantation/injection sites. Mice were deeply anesthetized with isofluorane and transcardially perfused with a 4% paraformaldehyde (PFA) solution prepared in phosphate-buffered saline (PBS). Extracted brains were postfixed in 4% PFA overnight at 4 °C, then kept in PBS until sectioning. Fixed brain tissue was sectioned using a microtome (Leica VT-1000) into coronal slices (thickness of 50–100 μm, depending on the experiment). Slices were stained with DAPI and mounted on glass microslides using Vectashield hardset mounting media. Mounted sections were stored at 4 °C until they were imaged using a laser scanning confocal microscope (Leica SP8).

Rabies viral vectors were made[72] by transfecting HEK 293T cells (ATCC CRL-11268) with expression vectors for the ribozyme-flanked viral genome (cSPBN-4GFP (Addgene 52487) or pRVΔG-4tdTomato (Addgene 52500), rabies viral genes (pCAG-B19N (Addgene 59924), pCAG-B19P (Addgene 59925), pCAG-B19G (Addgene 59921), and pCAG-B19L (Addgene 59922)), and the T7 polymerase (pCAG-T7Pol (Addgene 59926)). Supernatants were collected from 4 to 7 days after transfection, filtered, pooled, and passaged 3–4 times on BHK-B19G2 cells at a multiplicity of infection of 2–5. Supernatants were concentrated with ultracentrifugation, purified, and tittered[73].

To make AAV1-CAG-Flpo virus, the Flpo gene[74] was cloned into pAAV-CAG-FLEX-EGFP (Addgene 59331) to make pAAV-CAG-Flpo. Serotype 1 AAV was produced by triple transfection of HEK 293T cells with pAAV-syn-Flpo, pAAV-RC1, and pHelper (Cellbiolabs VPK-401) (per 15 cm plate, 15.5, 21.0, and 33.4 μg, respectively) using Xfect (Clontech 631318) transfection reagent. Supernatants and cells were harvested at 72 h post-transfection and viruses were purified and concentrated by iodixanol gradient centrifugation[75].

For quantification of dual-rabies retrograde tracing experiments from caudal and rostral ACC, we sliced the brain into 100 μm sections and imaged GFP and/or tdTomato fluorescence from every other slice using a ×10/0.4NA objective (Leica). Anatomical landmarks, such as position/size of ventricles, corpus callosum, striatum, and hippocampus, were used to manually align slices to a standard mouse brain atlas[46]. Back-labeled cells were counted using the cell-counter plugin in ImageJ (NIH). For the results presented in Fig. 1c, we counted the number of retrogradely neurons present in the different divisions of the VC and normalized by the total number of GFP- and tdTomato-expressing back-labeled cells found in the posterior cortex (0 to −4.0 mm posterior, relative to Bregma). For quantifying the distribution of cells projecting to the intermediate and deep layers of the SC (Fig. 1i), we counted the number of cells found in the ACC as a function of distance from Bregma in 0.5 mm bins. Since the sensory and intermediate/motor layers of the SC are <0.5 mm apart, it is challenging to restrict virus injection to the motor layer. However, the frontal cortex predominantly projects only to the motor layer; hence, to minimize contamination from virus spillover into the sensory layer of the SC, we normalized the number of neurons in the ACC by the total number of back-labeled neurons found in the frontal cortex (0 to +3 mm anterior, relative to Bregma).

We injected rabies viruses encoding GFP or tdTomato into the VC or the SC to identify ACC neurons projecting to these structures. We counted the total number of back-labeled neurons in the ACC (AP range 0 to 1 mm) every 200 μm in each animal and quantified the proportion of neurons that were labeled with GFP, tdTomato, or both out of all labeled cells.

We performed anterograde trans-synaptic tracing experiments[58] to identify SC neurons that receive inputs from the ACC using the AAV1-CAG-Flpo virus described above. We produced the Flp-dependent tdTomato reporter line Ai65F by crossing the Cre- and Flp-dependent tdTomato double-reporter line Ai65D[76] (Jackson Laboratory 021875) to the Cre deleter line Meox2-Cre[77] (Jackson Laboratory 003755), so that only Flp is required for expression of tdTomato. An AAV1-CAG-Flpo virus was injected in the ACC of these mice to label postsynaptic neurons with tdTomato. After allowing 4–6 weeks for Flpo and tdTomato expression, we sectioned the brain into 50 μm slices and used standard

immunohistochemistry techniques to identify GABA-expressing neurons co-labeled with tdTomato. The tissue was placed in 5% normal goat serum, 1% triton blocking solution in 0.1 M PBS for 1 h at room temperature. It was then incubated in primary antibodies against GABA and NeuN (rabbit anti-GABA, 1:500, A2052 Sigma; guinea pig anti-NeuN, 1:500, 266-004 Synaptic Systems) and 1% NGS/0.5% triton overnight at 4 °C. Following washing in 0.1 M PBS 3 × 10′, the tissue was incubated in secondary antibody (goat anti-rabbit IgG AlexaFluor 488, goat anti-guinea pig IgG AlexaFluor 647, 1:500; Invitrogen) and 1% NGS for 4 h at room temperature. Tissue was then washed 3 × 10′ in PBS, mounted, and coverslipped with anti-fade mounting medium (Prolong Gold; ThermoFisher). Tiled z-stacks were collected on a confocal microscope (SP8, Leica) using a ×20 objective at 1024 × 1024 resolution, 2 μm apart (~20 z-slices) from sections containing the SC (between −3.4 and −4.7 AP from bregma). A 1 × 1 mm ROI was selected over the intermediate and deep layers of SC and z-projected across 25 μm. Background was automatically subtracted (ImageJ) and NeuN+ cells were identified using an automated cell-counting binary mask (watershed segmentation, ImageJ). GABA+/NeuN+ and tDTom+/GABA±/NeuN+ cells were manually identified and calculated as the proportion of NeuN+ cells (cell-counter plugin, ImageJ).

**Behavioral apparatus and task training**. Mice were trained to report the spatial location of visual cues by rotating a trackball with their forepaws, similar to a previous design[10]. Animals were head fixed on a behavior rig assembled from optical hardware (Thorlabs), placed in a polypropylene tube to limit body movement, and positioned ~8 cm from an LCD screen (7″ diagonal; 700YV, Xenarc Direct) such that their forepaws rested on a trackball. Ball movements were monitored with a commercially available USB optical trackball mouse (Kensington Expert Mouse K64325). The original trackball was replaced with a 55 mm diameter ping pong ball (Joola), which was light enough for mice to rotate comfortably. We inserted a hypodermic tube down the center of the trackball so it could only be rotated along a single axis (left or right). To fix the ping pong ball to the optical mouse, we made grooves in the trackball chassis and secured the hypodermic needle with hot glue. A USB host shield (SainSmart) was used to connect the output of the optical mouse to a microcontroller (Arduino), which ran a custom routine that detected ball movements every 10 ms. In the event of a ball movement, the microcontroller outputted a timestamp and the amount of movement (in pixels) to a behavioral control computer (Dell). In addition, a timestamp was sent every 100 ms to synchronize timing between the microcontroller and the behavior computer. In our system, one pixel of optical mouse movement corresponded to ~0.15° movement on the ball. Behavioral control was implemented with custom software written in MATLAB (Mathworks) using the Psyhcophysics and Data Acquisition toolboxes.

During the inward task (Figs. 2–5), the presented visual cue (black square, ~20°) started on either the left or the right side of the LCD screen. The trackball controlled the location of the visual stimulus in closed loop in real time. The gain of coupling between the trackball and the stimulus was calibrated so that rotating the ball by the threshold amount (15°) in the correct direction moved the stimulus from its starting position to the center of the screen. Ball positions were accumulated throughout the trial until the stimulus reached the center or the response window expired. Under this closed-loop control, any movement in the incorrect direction displaced the stimulus farther away from the center and towards the edge of the screen. Hence, such movements had to be offset by additional movements in the correct direction for the stimulus to move to the center of the screen and for the trial to be considered correct. If the ball was moved opposite to the direction of the instructed cue by the threshold amount, the stimulus moved to the edge of screen and the trial was considered incorrect. The software operated similarly when mice were trained on the outward contingency task (Fig. 6), except that they were required to move the visual cue to the edge of the screen for the trial to be considered correct, and moving the stimulus to the center of the screen was considered an incorrect response. Water was given as reward on correct trials, which was delivered through a metal spout placed within the reach of the tongue; the amount dispensed was controlled by opening a solenoid valve for a calibrated period.

The following procedures were used to train mice on both the inward (Fig. 2a) and outward (Fig. 6a) task contingencies. Mice were taken through successive stages of training until they became proficient at the task. Once mice recovered from the surgery, they were water restricted for 5–7 days (≥1 mL/day) and then trained to lick a metal spout to obtain small water rewards (3–6 μL). If mice did not receive their water allotment during training, they were given the remaining amount as hydrogel (Clear H₂O) in their home cage. After mice reliably licked the waterspout, they earned water rewards by using the trackball to move the presented stimulus to the center of the screen. To discourage spontaneous trackball movements, mice were required to hold the ball still for 1 s to trigger trial start, which was signaled with an auditory tone (0.5 s, 1 kHz); the visual cue appeared with a 1 s delay after the onset of this tone. During early stages of training, only movements in the correct direction contributed to movement of the stimulus. Once mice reliably moved the ball in either direction on >90% of trials, this condition was removed, and the movement of the stimulus was fully coupled to the movement of the trackball. In the next stage of training, we used an anti-bias algorithm in which the same stimulus was repeated on consecutive trials if mice made an error until they performed the trial correctly; stimulus location on trials

following correct trials were randomly chosen. Once performance reached ~70%, the anti-bias algorithm was turned off and stimuli were presented in a randomized manner. Throughout all stages, auditory white noise was used to signal miss trials if mice failed to move the ball to threshold before expiration of the response window. As mice progressed through the training stages, we gradually decreased the response window from 10 to 1 s. Correct and incorrect trials were signaled with auditory tones (0.2 and 10 kHz, respectively), followed by an inter-trial delay of 2.5 s.

Animals trained for two-photon imaging experiments were taken through two additional stages of training. First, we turned off the visual feedback of the closed-loop coupling between the trackball and the stimulus, and instead flashed stimuli for 200 ms. This was to avoid the potential of evoking neural activity due to the movement of the visual cue on the screen. Second, we introduced uncertainty in temporal expectancy for stimulus onset by randomizing the period between the auditory cue signaling trial start and the onset of the visual stimulus (exponential distribution with mean of ~1.8 s, min and max delay of 1 and 5 s, respectively). Only mice trained for imaging experiments were taken through these steps.

A cohort of untrained mice were used to test the role of the SC in spontaneous trackball movements. Mice received an injection of AAV-syn-Jaws in the SC and implanted with an optic fiber cannula over the injection site, as described above. After mice recovered from surgery, they were water restricted for 5–7 days and trained to lick a metal spout to obtain small water rewards (3–6 μL). Once mice reliably licked the waterspout, we began sessions of randomly rewarded no-stimulus trials. During these sessions, no visual or auditory cues were presented that would signal the beginning or end of each trial. No-stimulus "trials" of 2 s duration were presented continuously, such that there was no delay between the end of one trial and the beginning of the next (note that this "trial" structure was not apparent to the mice and is only used for administering rewards, optogenetic stimulation, and subsequent analysis). On each trial, one movement direction was randomly selected by the software to be rewarded. If mice moved the ball in the designated direction during this period, they obtained a small water reward; no reward or feedback was provided for no-movement or "incorrect" trials (in which mice moved the ball to threshold, but in the unrewarded direction). To encourage movements in both directions, we simultaneously used two anti-bias algorithms during each session: (1) the movement direction selected for rewarding was repeated on consecutive trials until mice received a reward and (2) the rewarded movement direction switched once mice received a reward.

**Forepaw tracking**. We tracked the forepaws of freely moving mice during spontaneous turns and of head-fixed mice during ball rotations. Mice were placed in their home cage with the bedding removed. We placed an iPhone 11 camera below the cage to video record limb movements during locomotion. Videos were recorded at 30 frames per second with a 1280 × 720 pixel resolution. We visually identified individual leftward and rightward turns and extracted the corresponding frames as video clips (4–5 clips for each direction per mouse). We manually annotated positions of several body parts (left/right forepaws, the nose, and the base of the tail) in a subset of frames and used the DeepLabCut algorithm[52] to estimate their positions in the remaining frames. To define the position of the forepaws relative to the axis of the body, we first transformed the x–y positions of each tracked body part so that the base of the tail lies at the origin of a cartesian coordinate system. We then computed the angle between the base of the tail and the nose relative to the x-axis in each frame. This angle was used to generate a rotation matrix, which rotated the paws and the nose such that the nose position lies 90° to the base of the tail. For tracking in head-fixed mice, the camera was positioned to image the nose and paws of the animal on the ball. Clips of clockwise and counterclockwise ball rotations were manually extracted. We used DeepLabCut to track these body parts as well as the center of the headplate. The tracked positions were transformed similarly as above, except the headplate was placed at the origin of the coordinate system. We defined the body axis as the y-axis and computed the x-position of the paws in this coordinate system for both conditions. The values reported in Supplementary Fig. 2b, c are the mean x-position values of both paws, averaged over all frames of each video clip.

**Optogenetic manipulation of behavior**. Our behavioral paradigm involved both lateralized visual cues and lateralized actions. We refer to stimuli in the task as left and right cues. We define ball rotation actions as contraversive or ipsiversive (Fig. 2a, b and Supplementary Fig. 2) relative to the left hemisphere. Throughout the manuscript, we thus present all data assuming that recordings and inactivations were performed in the left hemisphere (see Table 1 for targeting details of specific experiments).

Photostimulation was provided with a solid state 593 nm laser for experiments using Jaws (OptoEngine). Laser stimulation was triggered from the behavior control computer and lasted from 0.3 s before to 1 s after visual cue onset. Across the mice, 30.4 ± 5.1% (mean ± standard deviation; range 13.3–34.5%) of trials were inactivated. Trials were randomly selected for inactivation, except photostimulation could not occur on two consecutive trials. The output of the laser was coupled to a patch cable (Thorlabs) with an FC/PC fiber coupler (OptoEngine). Laser power through the patch cable was measured with a digital power meter before each experiment (Thorlabs). In experiments requiring photostimulation through chronic windows over the VC, the ceramic ferrule of the patch cable was positioned

so it filled the entire 3 mm window and delivered a constant light pulse with 20 mW of power. In experiments requiring light delivery through implanted optical fiber, the ceramic ferrule of the patch cable was coupled to the fiber optic cannula with a ferrule mating sleeve (Thorlabs). Activity in the SC was inhibited by delivering 10 mW of constant yellow light. Photoinhibition of ACC outputs was achieved by constant yellow light (20 mW) illumination through an implanted fiber optic cannula (SC) or through a chronic window (VC).

For optogenetic manipulation of spontaneous movements in untrained mice, an average of 39% (±15%) of "trials" were pseudorandomly selected for photostimulation, with the additional stipulation that there be at least six consecutive trials (~12 s) between laser trials. Laser stimulation was triggered from the behavior control computer and lasted 1.3 s, beginning from 0.3 s before the (uncued) trial start.

**Behavioral analysis for optogenetic experiments**. We computed several metrics to quantify the performance on each trial type. The timeout rate is the proportion of trials in which the ball was not moved to threshold within the allotted response window out of all trials where a given stimulus was presented. The incorrect rate is the proportion of trials in which the ball was moved opposite the direction instructed by the cue out of all completed trials for each visual stimulus (i.e., excluding timeout trials). In other words, 100−incorrect rate equals the correct rate. The response time was the amount of time taken from stimulus onset to a complete correct response. The response window (i.e., time given for making a complete response) was set to 1 s. The mean instantaneous ball velocity was calculated from movement start (defined as when the ball was rotated by more than 0.5°) until a complete response was made. Side preference was quantified using Eq. (1), wherein contraversive and ipsiversive refer to the proportion of trials in which these actions were correctly selected (excluding timeout trials):

$$\frac{\text{contraversive} - \text{ipsiversive}}{\text{contraversive} + \text{ipsiversive}} \quad (1)$$

Inactivation-induced change in contraversive action bias shown in Figs. 5d and 6e was calculated with Eq. (2):

$$\Delta \text{ contra action} = \frac{(\text{contra}_{\text{corr,l}} + \text{ipsi}_{\text{incorr,l}}) - (\text{contra}_{\text{corr,nl}} + \text{ipsi}_{\text{incorr,nl}})}{(\text{contra}_{\text{corr,nl}} + \text{ipsi}_{\text{incorr,nl}})} \quad (2)$$

The right stimulus response bias in Fig. 6d was computed with Eq. (3) for both the inward and outward tasks:

$$\Delta \text{ right stimulus response} = \frac{(\text{right}_{\text{corr,l}} + \text{left}_{\text{incorr,l}}) - (\text{right}_{\text{corr,nl}} + \text{left}_{\text{incorr,nl}})}{(\text{right}_{\text{corr,nl}} + \text{left}_{\text{incorr,nl}})} \quad (3)$$

In the equations above, contra and ipsi refer to trials in which the contraversive and ipsiversive actions were cued, right and left refer to trials in which the right or left cues were presented, corr and incorr refer to correct and incorrect rates, and l and nl refer to laser and non-laser trials, respectively.

We quantified the effect of optogenetic inactivation on inward and outward tasks by comparing behavioral performance on non-laser and laser trials from the same sessions. Data from three optogenetic sessions were combined for each mouse. We used a permutation test to quantify the significance of changes in behavioral performance observed with optogenetic inactivation. We tested against the null hypothesis that the observed change in performance does not depend on laser inactivation. In each round of permutation, we randomly reassigned the laser labels across trials for each animal in an experiment. We then concatenated trials for all animals and recalculated the average laser-induced change in the incorrect rate, the timeout rate, and the response time for right cue and left cue trials. This process was repeated 1000 times, creating a distribution of performance changes expected by chance. The two-tailed $p$ value was computed as the proportion of performance changes that were as or more extreme than the observed change on either side of the distribution. We determined significance using the Bonferroni-adjusted alpha value of $0.05/2 = 0.025$ for analyses that compared the effect of inactivation on right and left cue trials.

We also tested the effect of optogenetic inactivation on spontaneous ball movements in untrained mice. Only trials with attempted movements, regardless of whether they were rewarded, were considered for analysis. Trials in which the ball was moved by the threshold amount of 5° were labeled as contraversive and ipsiversive based on the direction of movement and the SC hemisphere inactivated. Inactivation-induced changes in contraversive and ipsiversive movements, shown in Supplementary Fig. 5, were calculated with Eqs. (4) and (5), respectively, as follows:

$$\Delta \text{ contraversive} = \frac{(\text{contraversive}_{\text{l}} - \text{contraversive}_{\text{nl}})}{(\text{contraversive}_{\text{nl}})} \quad (4)$$

$$\Delta \text{ ipsiversive} = \frac{(\text{ipsiversive}_{\text{l}} - \text{ipsiversive}_{\text{nl}})}{(\text{ipsiversive}_{\text{nl}})} \quad (5)$$

In the equations above, l and nl refer to laser and non-laser conditions, respectively. We quantified the effect of optogenetic inactivation by comparing the proportion of movements of each type out of all attempted movements in laser and non-laser conditions. Inactivation was performed on left SC ($n = 2$ mice) and right

SC ($n = 3$ mice). Trials from four sessions were concatenated for each mouse. Significance testing was done using the permutation test described above.

**Two-photon microscopy**. GCaMP6s fluorescence was imaged through a 16×/0.8 NA objective (Nikon) using galvo-galvo scanning with a Prairie Ultima IV two-photon microscopy system. Excitation light at 910 nm was provided by a tunable Ti:Sapphire laser (Mai-Tai eHP, Spectra-Physics) equipped with dispersion compensation (DeepSee, Spectra-Physics). Emitted light was collected with GaAsP photomultiplier tubes (PMTs; Hamamatsu). A blackout curtain was attached to a custom stainless-steel plate (eMachineShop.com), which was mounted on the headplate to prevent light from the LCD screen from entering the PMTs. Layer 5 GCaMP6s-expressing neurons in transgenic animals were imaged with 2× optical zoom, 350–550 μm below the brain surface for 20-min-long behavioral sessions. Upon completion, excitation wavelength was changed to 830 nm, which produced both a signal from the retrobeads and a structural green signal from GCaMP6. Emission signals were split with a dichroic mirror (FF649-Di01; Semrock) mounted in a filter cube and directed to two different PMTs. Simultaneous imaging of both signals with 830 nm excitation facilitated spatial alignment of retrobead signals to GCaMP6s signals collected during behavior and allowed us to identify SC-projecting ACC neurons (see "Image analysis"). 4× optical zoom was used for imaging callosal or VC axons in the ACC up to 100 μm below the surface. All imaging experiments were performed at a frame rate of 5 Hz. Laser power at the specimen was controlled with pockel cells and ranged from 10 to 50 mW, depending on GCaMP6 expression levels and depth.

**Image analysis**. Images were acquired using the PrairieView software (Bruker) and saved as multipage TIFF files using ImageJ (NIH). Image processing and region of interest (ROI) selection was performed in ImageJ (NIH). To correct for lateralized movements in the $x$–$y$ axis, images were realigned to a reference frame (the pixel-wise mean of all frames) using the template matching plugin[78]. ROIs were drawn manually over visually identified neuronal somas. We used reference frames generated by taking the pixel-wise maximum, mean, and standard deviation projection of all frames. These projections were compared with each other to visually identify neurons. In most cases, neurons were visible in all three projections; however, some neurons could be more readily identified in one of the three projections. For example, highly active neurons with a low baseline fluorescence could be difficult to identify in the mean projection but show up well in the standard deviation projection. Neurons had to be visually present in at least one of these projections to be included. We additionally verified that the visually selected neuronal soma had dynamic GCaMP6s signals. We defined the neuropil for each neuron by placing an ROI with the same shape as the somatic ROI over an adjacent area devoid of other neurons. To minimize the contribution of the neuropil signal to the somatic signal, corrected neuronal fluorescence time series was estimated with Eq. (6)[79]:

$$F = F_{\text{raw\_soma}} - 0.7\,(F_{\text{raw\_neuropil}}) \quad (6)$$

Similar image analysis was used for axonal imaging experiments, except ROIs were drawn over visually identified boutons and fluorescence signals were not adjusted for neuropil contamination. $\Delta F/F$ (DFF) for each neuron or bouton was calculated with Eq. (7):

$$\text{DFF} = (F - F_0)/F_0 \times 100 \quad (7)$$

In Eq. (7), $F_0$ was the fluorescence value with the highest density (estimated using MATLAB function ksdensity). To correct for drifts in baseline, we employed the method described in Pachitariu et al.[80]. We computed a moving baseline by filtering the data with a Gaussian of 1 s width, then minimum filtering followed by maximum filtering with a window of 60 s. The resulting baseline was subtracted from the DFF trace before $z$-scoring.

To identify retrobead-containing neurons, a reference frame was generated by taking the pixel-wise mean of realigned GCaMP6s frames acquired during the behavioral session at 910 nm. The green channel acquired with excitation at 830 nm was realigned to this reference using the template matching plugin. The resulting translation values were then applied to the channel containing signals from retrobeads. An average projection was taken after this realignment and superimposed onto the 910 nm GCaMP reference frame used for drawing ROIs. Neurons containing retrobeads were then visually identified.

**Analysis of visual activity in axons**. We assayed the visual responsiveness of VC or callosal axons in the ACC in awake animals passively viewing visual stimuli presented at the same locations as during the task (solid black square for VC axons, gratings drifting at 90° for callosal axons; 1 s stimulus duration). Visually responsive boutons were identified by comparing pre-stimulus activity averaged over a 1 s period to averaged activity 0.6–1.2 s after stimulus onset (two-sided Wilcoxon signed-rank test, $p < 0.01$). AUROC values were computed for each visually responsive bouton using stimulus activity on contralateral and ipsilateral cue trials. A preference score was then computed with Eq. (8):

$$\text{Preference score} = 2(\text{AUROC} - 0.5) \quad (8)$$

This score ranged from 1 (complete contra preference) to −1 (complete ipsilateral preference). To determine if individual boutons had significant stimulus

preference, scores were recomputed after shuffling trial labels 1000 times. Observed preference scores outside the center 95% of the shuffled distribution were considered significant ($p < 0.05$, two sided).

**Analysis of task responses of ACC neurons.** For imaging experiments during the inward contingency task (Figs. 2 and 3), the session-wide DFF trace for each neuron was z-score normalized. Responses on individual trials were aligned to visual stimulus onset. Trials were labeled as right cue-contraversive action, right cue-ipsiversive action, left cue-contraversive action, and left cue-ipsiversive action, with action specified relative to the left hemisphere. We only analyzed sessions with at least five trials in each condition (note that the right cue-ipsiversive and left cue-contraversive conditions correspond to incorrect trials). The color plots in Figs. 2f and 3c were generated by averaging responses on each trial condition for each neuron. Note that rows correspond to neurons, and each row plots activity of the same neurons across the four conditions. For analyses in Fig. 2g, i and 3d, responses were averaged over a 1-s post-stimulus period (post-stimulus response) and then compared for the indicated trial conditions using Wilcoxon signed-rank tests. To determine trial activity on right cue trials (Fig. 2i), we first separately averaged post-stimulus responses on right cue-contraversive action and right cue-ipsiversive action trials; responses on these trial types were then averaged together to generate right cue responses. Trial activity on left cue, contraversive action, and ipsiversive action trials was determined similarly, except post-stimulus responses on the following trial types were used, respectively: (1) left cue-contraversive action and left cue-ipsiversive action; (2) right cue-contraversive action and left cue-contraversive action; and (3) right cue-ipsiversive action and left cue-ipsiversive action.

**Decoding actions from neuronal activity.** We built linear support vector machine (SVM) classifiers using the LIBSVM library for MATLAB[81] to test whether the activity of ACC-SC neurons can be used to predict which action was selected by the animal. Value of $10^{-4}$ was used for the regularization parameter C. We performed two types of decoding analyses. For both analyses, task responses were aligned to the response time and single-trial activity was averaged over a window of $-0.2$ s before to 0.4 s after the response. In the first analysis (Fig. 3f), decoding was performed with ACC-SC neurons recorded across all eight behavioral sessions. We constructed "pseudotrials" by combining neuronal responses recorded across different sessions. SVM classifiers were trained to predict contraversive or ipsiversive trials based on the activity of individual trials. Activity for the contraversive and ipsiversive action label was taken from equal numbers of right and left cue trials. We ran 1000 iterations of the model. There was an unequal number of trials between conditions, so we used subsampling to balance trial types on each iteration (5 trials from each condition). We minimized model over-fitting by using the cross-validation technique (10-fold) to split the data into a training and testing set on each iteration. Classifier performance on each iteration was estimated by averaging prediction accuracies across the 10 splits. Final classifier accuracy was determined by averaging these mean accuracies across all iterations. To determine if the action decoder performed above chance, we used an identical procedure except we shuffled labels for the test data. Mean prediction accuracy derived from correctly labeled test data and falling outside the center 95% of the shuffled distribution was considered significant.

In the second decoding analysis (Fig. 3g), we followed an identical procedure except that separate classifiers were trained for each recording session. Moreover, we also trained classifiers with the activity of unlabeled ACC neurons. We had less ACC-SC neurons than unlabeled neuron in each session. Hence, we randomly subsampled to match the number of unlabeled neurons to the available number of ACC-SC for each iteration. The decoding accuracy was estimated for each session separately by averaging across the 1000 iterations. Single session decoding accuracy for the two neuronal populations was compared using a Wilcoxon signed-rank test.

**Electrophysiological recordings in the SC.** For photostimulation of the ACC, we injected AAV1.CaMKII.hChR2(H134R)-mCherry (University of Pennsylvania vector core) at coordinates AP: 0.5 mm, ML: 0.5 mm, and DV: 0.4 and 0.9 mm (250 nL at each site). Cannulas were implanted such that the optical fiber was 0.3 mm below the pia (300 μm/0.39 NA core fiber optic coupled to a 2.5 mm stainless-steel ferrule; CFMC13L02; Thorlabs). We tested the effect of photostimulating ChR2-expressing ACC neurons on activity in the SC using a 473 nm blue solid state laser (Optoengine). One or two days before the experiments, mice were habituated to head fixation in 1 h sessions. On the day of the experiment, mice were anesthetized with isoflurane and the Metabond and silicone elastomer placed during the initial surgery were removed from the skull. The mouse was placed on the stereotaxic frame and a 500 μm diameter craniotomy was performed on top of the recording site (from bregma: $-3.6$ to $-4$ mm anteroposterior and 0.8 to 1 mm mediolateral). The dura above the cortex was removed and the craniotomy was protected with saline and a piece of gelfoam. SC craniotomy was performed on the same side as that implanted with the fiber optic cannula over the ACC. The skull was covered again with silicone elastomer and the animal was returned to its home cage to recover from anesthesia for at least 2 h. After recovery, mice were head fixed and the silicone and gelfoam overlaying the craniotomy was gently removed. 0.9% NaCl solution was used to keep the surface of the brain wet for the duration of the recordings.

After placing the animal in the recording set up, we submerged a reference silver wire in the NaCl solution on the skull surface. The position of the 16-channel silicone probe (A1×16-Poly2-5 mm −50s-177-A16, NeuroNexus) was referenced on lambda and the surface of the brain and lowered slowly (1 min per mm) to reach superficial sensory layer of the SC (~1.3 mm in the ventral axis) using a motorized micromanipulator (MP –285; Sutter Instrument Company). The extracellular signal was amplified using a 1× gain headstage (model E2a; Plexon) connected to a 50× preamp (PBX-247; Plexon) and digitalized at 50 kHz. The signal was highpass filtered at 300 Hz. Once the visual layer of the SC was identified (characterized by strong and reliable visual responses to drifting gratings and sparse noise), the recording probe was lowered ~400 μm deeper to the motor layer of the SC. For successful recordings, the silicone probe was gently retracted and the recording tract was marked by re-entering with DiI coated probe (2 mg/mL—D3911, ThermoFisher Scientific) at the same location. For some experiments, we were able to record from two locations spaced 500 μm apart in the dorsal–ventral axis. The brain was harvested post hoc and sectioned to confirm the probe location and ChR2 expression in the ACC. Spikes were isolated online with amplitude threshold using Plexon Recorder software, but re-sorted using the MountainSort (v3) automated spike sorting algorithm[82].

Units were curated manually after automatic detection using the following criteria. Single units sorted from the MountainSort algorithm were visualized with the MountainView software for manual curation. We first made a visual inspection to remove artifacts, which typically included large spikes observable in most channels (noise units) and symmetrical spike waveform of low amplitude with high firing frequency. Using spike auto-correlograms, we rejected all units with a large number of spikes happening in the refractory period of another spike (more than 0.25% of spikes occurring at <1 ms after a spike event). By examining the amplitude histogram of each unit, we removed units with cropped amplitude distribution that could have resulted from thresholding. We finally merged single units that were split into two units but had similar waveforms. To do so, we inspected their time series and merged two similar units that resulted from drift from the probe. In case of doubt whether two similar units were from the same neuron, we excluded them. For each resulting unit, we use only the portion of the recording where the baseline firing rate was constant.

Visual stimuli were presented during recordings in the SC to increase neuronal responsiveness. Sparse noise on a 3 × 5 grid (square size was the same as used for behavior experiments) of black and white square on a gray background (50% luminance) were displayed for 0.1 s, followed by a 0.1 s gray screen period. Positions were randomized within each block such that black and white squares were presented once at each of the 15 positions. The total duration of a block was 6 s, with 1 s inter-block intervals. Photostimulation of the ACC (10 ms blue light pulses at 20 Hz) was performed on 50% of the blocks. Photostimulation started 0.5 s before visual stimulus presentation and was turned off 0.5 s after stimulus offset (total duration of 7 s).

Since ACC axons in the SC target the intermediate and deep layers, we focused our analysis on recordings made from these areas. While we observed robust retinotopically organized visual responses in the superficial layer, we rarely encountered cells in the deeper layers that specifically responded to the location of sparse noise stimuli. Therefore, recordings from deeper neurons likely reflect ongoing, spontaneous activity that is modulated by the visual stimulation. We first determined if individual neurons were significantly modulated by laser activation of the ACC by comparing the firing rates (FR) of activity on non-laser and laser trials. For each modulated neuron, we also computed a laser modulation index with Eq. (9):

$$\frac{\mathrm{FR_{Laser}} - \mathrm{FR_{nonlaser}}}{\mathrm{FR_{Laser}} + \mathrm{FR_{nonlaser}}} \qquad (9)$$

**Reporting summary.** Further information on research design is available in the Nature Research Reporting Summary linked to this article.

## Data availability
The data that support the findings of this study are available from the corresponding authors upon reasonable request. Source data are provided with this paper.

## Code availability
Custom code used in this work is available from the corresponding authors upon reasonable request.

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

## Acknowledgements

We thank Michael Halassa and Diana Poon for thoughtful discussions and Vincent Pham for assistance with histology. This work was supported by grants from National Eye Institute F32 EY024857 to R.H., National Institute of Mental Health K99 MH112855 to R.H., National Eye Institute F32 EY028028 to G.O.S., Natural Sciences and Engineering Research Council of Canada PDF 487824-2016 to V.B.-P., National Eye Institute F31 EY031259 to K.G.C., National Institute of Mental Health U01 MH106018 and U01 MH109129 to I.R.W., National Eye Institute R01 EY028219 and R01 EY007023 to M.S., National Institute of Neurological Disease and Stroke U01 NS090473 to M.S., National Science Foundation EF1451125 to M.S., and Simons Foundation Autism Research Initiative to M.S.

## Author contributions

R.H. and M.S. conceived the project and developed the concepts presented. R.H. designed the experiments and analysis approach with contribution from M.S. R.H. and G.N.P. designed and implemented the behavioral task. R.H. performed and analyzed the anatomical, behavioral, optogenetic, and calcium imaging experiments with contributions from G.O.S., K.G.C., L.M.G., A.S., and E.A. V.B.-P. performed and analyzed the extracellular recording experiments with input from R.H. I.R.W. provided unpublished reagents. R.H. and M.S. wrote the manuscript with input from all authors.

## Competing interests

The authors declare no competing interests.
