## [Peer Review File · Nature Communications]

Reviewers' Comments:

Reviewer #1:

Remarks to the Author:

This manuscript by Huda et al. investigates the top-down influence of the anterior cingulate cortex (ACC), a part of the prefrontal cortex (PFC), on sensorimotor function, using an elegant visually guided two-choice behavioral task, projection-specific two-photon calcium imaging and optogenetic manipulations. The main findings are: 1) ACC integrates visual information of the contralateral field from the visual cortex (VC) and that of the ipsilateral field from the callosal input, and in turn projects to the superior colliculus (SC) on the same side; 2) the ACC to SC output promotes clockwise actions, which counteracts an intrinsic counterclockwise response bias of SC neurons; 3) the ACC to VC output facilitates sensory processing of contralateral visual cues; 4) ACC-VC and ACC-SC neurons are distinct subpopulations. Using opposite cue-response contingencies, the authors were able to dissociate the sensory and motor components of sensorimotor control, which has an advantage over the previous delay-response strategies. Overall, the results are exciting and data in general support the conclusions. I have only some specific points to improve the manuscript.

Specific points:

1. Page 4, line 86, "Although both compartments received inputs from medial higher visual cortex.....". The anatomical data also show significant input from the lateral higher visual cortex (Figure 1B). In fact, the labeled cell density is comparable between different cortical areas. The summary in Figure 1C appears inconsistent with the raw image shown in Figure 1B. Please clarify in the figure legend or text which side of the brain was imaged. A diagram of wiring between VC, ACC and SC would be helpful.
2. In Figure 1B, please mark cortical layers.
3. Comparing Figure 4B and Supplementary Figure 5B, the effect of silencing VC is surprisingly small. Does this mean that this behavioral performance is largely independent of VC? Please add discussion on this issue.

Reviewer #2:

Remarks to the Author:

In this experimental tour-de-force, Huda and colleagues use a combination of anatomical tracing, GCaMP imaging and optogenetic perturbations with and without projection specificity to investigate the role of cingulate projections to the superior colliculus and the visual cortex in a visual sensorimotor task. Their findings suggest that caudal ACC to SC projections are involved in action selection, but ACC to visual cortex projections enhance sensory processing. While specifically the latter finding is not novel (e.g. Zhang et al, Science, 2014), the manuscript as a whole significantly adds to our understanding of how different frontal projections help coordinate different aspects of sensorimotor behavior. The question is timely and relevant, the methods are sound and most results are presently clearly. However, as I detail below, further analyses are warranted to better support some of the central claims, and help with the interpretation of the results. In particular, it is presently difficult to reconcile their findings of SC promotion of counterclockwise movements with a long SC literature showing its involvement in contralateral orienting movements. For this reason, I think it is crucial to understand in which ways the head-fixed ball turning motor output is similar to or differs from more traditional readouts, and how those specific kinematics impact neural activity. Of course, the possibility remains that ball-turning movements are fundamentally different than orienting responses, but it would be important to show this more convincingly.

Major issues

Clockwise vs. counterclockwise movements. I would suggest changing this to more usual contra- and

ipsi-lateral nomenclature. Perhaps it is just me, but this required a lot of mental gymnastics to understand the laterality of the effects, since clockwise means different things for right and left stimulus trials. Because of this reason, grouping left- and right-hemisphere data by flipping CW / CCW labels is misleading. Moreover, it is presently unclear whether an inward movement in a left trial actually translates to literal clockwise rotations of the trackball. At the very least, I would suggest expanding the explanation of the relationship between CW / CCW and the more traditional orienting movements in the beginning of the results section, to help orient the reader.

Towards the end of the discussion (line 322), the authors do touch upon the rotation vs orienting movement comparison, and suggest a relationship between inward / outward contingencies and pro / anti tasks, which could indeed explain some of the findings. Here, a video or EMG analysis of limb / neck movements would provide much stronger support for this claim. Moreover, if this is indeed the case, one might expect that task performance in the outward contingency is worse than inward, as is the case for anti as compared to pro contingencies. Is this the case?

More generally, it would be crucial to add quantification of the trackball movements themselves (i.e. optical sensor readout). This would help clarify what is the relationship between movement on the screen and the trackball. This should be done both for control trials, and in all instances of optogenetic manipulations, e.g. by plotting laser-triggered movement trajectories. This would have the added benefit of showing that the manipulations specifically impact chosen movement direction (i.e. the decision), rather than lower-level kinematics.

Behavioral quantification, Fig. 2B. In addition to kinematic analysis, please also show a quantification of side biases, i.e. the difference in performance between ipsi and contra trials. Given that laterality of behavior and neural activity is central to the main claims, it is crucial to show that the findings are not trivially related to intrinsic side preferences the animals may have.

Quantification of GCaMP activity, Figs. 2 and 3. Given the long decay time constants of GCaMP6s, as well as the fact that multiple behavioral events co-occur within a trial, simply doing trial-type averaging is potentially problematic. For example, for the neuron show in Fig. 2B, it is arguable that the stimulus response itself (i.e. $< \sim .5$ s) is not different between trial types. Although it helps that only the 1-s response window was analyzed, it is still important to show that the modulation of visual responses by action is not due to (kinematic) aspects of the motor output other than the direction of turn. I believe a regression-type approach to quantify neural activity as a function of all behavioral events is warranted here. Moreover, I would suggest plotting the average movement trajectories alongside with the activity plots.

Moreover, in addition to quantifying the average magnitude of activity across trials (or even better, regression coefficients), the authors should also quantify the proportions of neurons that significantly respond to the different stimulus-action combinations.

Fig 3. To strengthen the claim of the relevance of this ACC>SC projection, it would be important to show that action encoding is overrepresented in this population as compared to the overall population of ACC neurons (i.e. Fig 2). In Fig. 3E, are response magnitudes higher in this population as compared to the data in Fig. 2? What about proportions of neurons that significantly encode action? Is the action decoding accuracy from ACC>SC neurons (Fig. 3G) higher than the general ACC population?

SC recordings, Fig. S4. This is important data to help interpret the results. It would be very interesting to further analyze the effects of ACC activation in the initial parts of the laser ON epoch (~ 1 s), matching the response window during the task. In particular, are there any differences in the prevalence, magnitude and time course of inhibited vs excited units in the SC during this initial period? This would get at the possibility that e.g. an early net inhibitory effect is followed by more prolonged excitation.

Minor issues:

Throughout the manuscript, the authors refer to ACC projections to motor layers of the SC as top-

down input (e.g. 16, 54 etc). However, I would argue that “top-down input” is usually assumed to be comprised of feedback projections within a hierarchy, and is typically used to describe cortico-cortical connections. The author’s own data suggest, however, that the SC is a key node in the generation of the motor output itself, which many would argue would place it higher in a hierarchical scheme implementing sensorimotor transformations. To avoid confusion in readers, I would suggest replacing top-down input by simply input.

Rabies virus tracing in rostral and caudal ACC, line 85. It would help to add details here to help readers understand the strategy. Even after reading the methods, I am not entirely clear on how this experiment was setup. For instance, what mouse line was used? Do I understand correctly that this was an AAV with a CAG promoter injected in wild-type mice? If so, this would imply no cell-type specificity in the target regions, and if that is the case, why go for RV in the first place? To be clear, none of these concerns call into question the results, but please clarify the methodology.

Laser effect panels in Figs 4 - 6. Please add more ticks to the y axis, it is presently difficult to visually estimate effect sizes.

Methods, line 763. For the sake of reproducibility, I would suggest spelling out the laser probability (or range thereof), rather than saying ~20%. Also, what does pseudorandom mean here, exactly? Is it balancing contra and ipsi trials? Please clarify.

Methods, line 830. Most statistical tests in the manuscript used a permutation strategy, which is sound. Is there a reason why this one is a simple t-test?

Methods, ROI selection, lines 858-865. More details are needed here. For example, what does it mean that maximum, mean and standard deviation projections were all used? Did an ROI have to present in all three to be counted? How is the neuropil area defined, e.g. is it a ring of n pixels around the ROI? Finally, was F0 a constant value throughout the session or was it computed over a sliding window? If the former, was there any other method apply to correct for baseline drifts?

Methods, line 955. What criteria were used in manual spike cluster curation?

Response to reviewers

We thank the editor and the reviewers for their constructive feedback on our manuscript. We have performed additional experiments and analyses which we believe address all reviewer concerns. Below, we address each comment while noting changes/additions that incorporate reviewer feedback in the text and the figures. Note that reviewer comments appear in black and our responses are in blue.

Reviewer #1 (Remarks to the Author):

This manuscript by Huda et al. investigates the top-down influence of the anterior cingulate cortex (ACC), a part of the prefrontal cortex (PFC), on sensorimotor function, using an elegant visually guided two-choice behavioral task, projection-specific two-photon calcium imaging and optogenetic manipulations. The main findings are: 1) ACC integrates visual information of the contralateral field from the visual cortex (VC) and that of the ipsilateral field from the callosal input, and in turn projects to the superior colliculus (SC) on the same side; 2) the ACC to SC output promotes clockwise actions, which counteracts an intrinsic counterclockwise response bias of SC neurons; 3) the ACC to VC output facilitates sensory processing of contralateral visual cues; 4) ACC-VC and ACC-SC neurons are distinct subpopulations. Using opposite cue-response contingencies, the authors were able to dissociate the sensory and motor components of sensorimotor control, which has an advantage over the previous delay-response strategies. Overall, the results are exciting and data in general support the conclusions. I have only some specific points to improve the manuscript.

We thank the reviewer for their positive comments and constructive feedback on our manuscript. Below we address each comment.

Specific points:

1. Page 4, line 86, “Although both compartments received inputs from medial higher visual cortex.....”. The anatomical data also show significant input from the lateral higher visual cortex (Figure 1B). In fact, the labeled cell density is comparable between different cortical areas. The summary in Figure 1C appears inconsistent with the raw image shown in Figure 1B. Please clarify in the figure legend or text which side of the brain was imaged. A diagram of wiring between VC, ACC and SC would be helpful.

We agree with the reviewer and have modified the above statement to the following (lines 85-88): “Although both compartments received inputs from medial higher visual cortex, corresponding to functionally-defined anteromedial and posteromedial areas⁵¹, as well as the lateral higher visual cortex...”

As for comparing the summary results in Figure 1C with the raw image shown in Figure 1B, we previously selected an example histological section with sufficient retrograde labeling in all three VC areas (medial and lateral higher VC and primary VC) for illustration purposes. We agree with the reviewer that this is not a representative example reflecting summary statistics. To avoid

confusing readers, we have substituted Figure 1B with a different histological section from the same animal that better represents the summary results.

We have clarified in the figure legend that retrograde labeling was observed in the left VC, which is the same side as the rabies virus injection in the caudal and rostral ACC. We have also included a schematic diagram summarizing these tracing results in Figure 1J.

2. In Figure 1B, please mark cortical layers.

We have included markers for cortical layers in Figure 1B.

3. Comparing Figure 4B and Supplementary Figure 5B, the effect of silencing VC is surprisingly small. Does this mean that this behavioral performance is largely independent of VC? Please add discussion on this issue.

We thank the reviewer for bringing up this important point. We have added the following to the discussion in lines 370-379 to address this comment:

“Surprisingly, we found that VC inactivation only modestly affected behavioral performance, decreasing correct responding and increasing the timeout rate on right cue trials (Supplementary Figure 7). Inactivation of the SC led to a more robust behavioral effect (Figure 4B). A parsimonious interpretation of these findings is that focal viral inactivation of the VC, potentially targeting a subset of output pathways, reveals a relatively small contribution to performance in this task. At the same time, we found that the ACC-VC pathway facilitates responses to specific visual cues, demonstrating that ACC modulation of the VC is an important contributor to task performance. In future experiments, it will be important to use transgenic mice stably expressing opsins for large areal inactivation and projection-specific targeting of various VC output projections to test the full measure of VC contributions.”

Reviewer #2 (Remarks to the Author):

In this experimental tour-de-force, Huda and colleagues use a combination of anatomical tracing, GCaMP imaging and optogenetic perturbations with and without projection specificity to investigate the role of cingulate projections to the superior colliculus and the visual cortex in a visual sensorimotor task. Their findings suggest that caudal ACC to SC projections are involved in action selection, but ACC to visual cortex projections enhance sensory processing. While specifically the latter finding is not novel (e.g. Zhang et al, Science, 2014), the manuscript as a whole significantly adds to our understanding of how different frontal projections help coordinate different aspects of sensorimotor behavior. The question is timely and relevant, the methods are sound and most results are presently clearly. However, as I detail below, further analyses are warranted to better support some of the central claims, and help with the interpretation of the results.

We thank the reviewer for their insightful comments and constructive feedback on our manuscript. We have addressed all reviewer comments through additional analyses and experiments and provide a point-by-point response below. Please note that we have numbered reviewer comments.

In particular, it is presently difficult to reconcile their findings of SC promotion of counterclockwise movements with a long SC literature showing its involvement in contralateral orienting movements. For this reason, I think it is crucial to understand in which ways the head-fixed ball turning motor output is similar to or differs from more traditional readouts, and how those specific kinematics impact neural activity. Of course, the possibility remains that ball-turning movements are fundamentally different than orienting responses, but it would be important to show this more convincingly.

We thank the reviewer for bringing up this important issue, for which we have performed additional experiments and analyses (Supplementary Figure 2). Our interpretation is that head-fixed ball rotation movements are akin to orienting or turning movements in freely moving mice, which are known to be controlled by SC activity. We propose that counterclockwise ball rotations are rightward turning and clockwise rotations are leftward turning. We come to this conclusion based on two key similarities between the head-fixed and freely moving conditions.

A counterclockwise rotation of the ball in response to a right cue, for example, brings the stimulus to the center of the visual field in the 'inward' task. During free movement, rightward turning similarly allows the animal to orient to a right visual cue and bring it to the center of the visual field (Figure 2B). Hence, from an egocentric perspective, counterclockwise rotation movements are analogous to rightward orienting movements (and vice versa for clockwise rotations).

Another important similarity is in the movement of the forelimbs. We acquired new video data from freely moving mice during spontaneous turning behavior and head-fixed mice rotating the ball. We tracked the forelimbs in either setting using DeepLabCut and quantified the position of the paws relative to the axis of the body (Supplementary Figure 2; lines 115-124; lines 608-627). During freely moving turning, the forepaws were positioned opposite to the turning direction. For right turns, average paw position was to the left of the body, whereas for left turns, paws were instead positioned to the right (Supplementary Figure 2A, B). During counterclockwise ball turning in head-fixed mice, the paws were positioned to the left of the body, as expected for rightward turning. Clockwise ball rotations were associated with the opposite paw positions, as expected for leftward turning (Supplementary Figure 2C). This new analysis thus confirms that ball rotations in head-fixed mice are analogous to orienting responses in freely moving mice.

Importantly, our results with SC inactivation are also consistent with the view that ball movements are akin to orienting responses. We found that left SC inactivation during the task reduced counterclockwise movements (Fig. 4B), as would be expected for rightward orienting. Importantly, we also inactivated the SC in mice not trained on any specific sensorimotor contingency and found that left SC inactivation reduced contraversive (counterclockwise) ball actions (Supplementary Figure 5).

Together, our results show that ball movements in this task tap into the intrinsic orienting system of the animal and establish ball rotations as analogous to orienting movements (also see response to points 1 and 2 below).

Major issues

1) Clockwise vs. counterclockwise movements. I would suggest changing this to more usual contra- and ipsi-lateral nomenclature. Perhaps it is just me, but this required a lot of mental gymnastics to understand the laterality of the effects, since clockwise means different things for right and left stimulus trials. Because of this reason, grouping left- and right-hemisphere data by flipping CW / CCW labels is misleading.

We had originally labeled ball movements as counterclockwise/clockwise to describe responses from the perspective of how the ball is rotated. However, we agree with the reviewer that presenting these results from the egocentric perspective of the animal allows both a clearer description of our experimental results as well as reconciliation with the existing SC literature on orienting. Importantly, our new forepaw tracking analysis suggests that ball rotations in head-fixed mice are akin to orienting movements in freely moving mice.

We have followed the reviewer's suggestion regarding nomenclature as closely as possible. Our behavioral paradigm uses both lateralized cues and actions. Cue locations are in the right or left visual field, and actions are defined with respect to the left SC as contraversive and ipsiversive (Figure 2B, Supplementary Figure 2). For a cue in the right visual field, an inward action bringing it to the center is contraversive (in short, contra), consistent with the orienting movement a freely moving animal would make towards the cue. For a cue in the left visual field, an inward action is ipsiversive (in short, ipsi).

Moreover, it is presently unclear whether an inward movement in a left trial actually translates to literal clockwise rotations of the trackball.

It does, given that the ball was only allowed to move in one axis. A clockwise rotation of the trackball on a left cue trial produces inward movement of the cue (i.e., from its starting peripheral location to the center of the screen). The same rotation on a right trial moves the cue outward on the screen, which is an incorrect response in the 'inward task'. The coupling between cue movement on the screen and ball movements are the same in the 'outward task', except that the animal is rewarded for moving the cue outward to the outside of the screen instead of the center.

At the very least, I would suggest expanding the explanation of the relationship between CW / CCW and the more traditional orienting movements in the beginning of the results section, to help orient the reader.

In light of our new analyses, we have modified the initial description of our task and added more explanation to link ball rotations with orienting movements, as described above (lines 115 – 130; Figure 2B). We have removed the CCW/CW nomenclature, and have relabeled actions as contraversive and ipsiversive. In Fig 2 and all subsequent figures, we now clearly identify cue locations (left/right) and actions (contraversive/ipsiversive).

2) Towards the end of the discussion (line 322), the authors do touch upon the rotation vs orienting movement comparison, and suggest a relationship between inward / outward contingencies and pro / anti tasks, which could indeed explain some of the findings. Here, a video or EMG analysis of limb / neck movements would provide much stronger support for this claim. Moreover, if this

is indeed the case, one might expect that task performance in the outward contingency is worse than inward, as is the case for anti as compared to pro contingencies. Is this the case?

We thank the reviewer for this excellent suggestion. We present results from our new video analysis demonstrating the similarities between ball movements and turning movements (Supplementary Figure 2; Figure 2B; also see response to summary above). To test for differences in behavioral performance between the two tasks, we compared performance accuracy on the outward task (Figure 6) and the light control experiment during the inward task (Supplementary Figure 4B). We found that accuracy was indeed significantly better on the inward than outward task (inward: $91 \pm 3\%$; outward: $80 \pm 2\%$; $p < 0.05$).

3) More generally, it would be crucial to add quantification of the trackball movements themselves (i.e. optical sensor readout). This would help clarify what is the relationship between movement on the screen and the trackball. This should be done both for control trials, and in all instances of optogenetic manipulations, e.g. by plotting laser-triggered movement trajectories. This would have the added benefit of showing that the manipulations specifically impact chosen movement direction (i.e. the decision), rather than lower-level kinematics.

We include a new supplementary figure (Supplementary Figure 8) with the suggested analysis. We plot ball position trajectories for contraversive and ipsiversive responses during control and laser trials aligned to visual cue onset. The time window of $-0.3s$ to $+1s$ around cue onset covers the duration of the photostimulation. Overall, we find that ball movement trajectories during control and laser trials are largely similar for ACC projection manipulations. In the experiment with SC inactivation, contraversive trajectories are modestly affected. However, a potential caveat in interpreting this effect is that inactivation produces very few contraversive responses (Fig. 4B). We report these results in the text in lines 289-292.

4) Behavioral quantification, Fig. 2B. In addition to kinematic analysis, please also show a quantification of side biases, i.e. the difference in performance between ipsi and contra trials. Given that laterality of behavior and neural activity is central to the main claims, it is crucial to show that the findings are not trivially related to intrinsic side preferences the animals may have.

We now include an additional panel in Fig. 2C showing the distribution of side preference across mice. Importantly, mice do not have a significant side preference (mean side preference 0.03 ± 0.01). We have also added the following to the methods (lines 663-665): “Side preference was quantified using the following equation, wherein contraversive and ipsiversive refer to the proportion of trials in which these actions were correctly selected (excluding timeout trials): $\frac{\text{contraversive} - \text{ipsiversive}}{\text{contraversive} + \text{ipsiversive}}$ ”

5) Quantification of GCaMP activity, Figs. 2 and 3. Given the long decay time constants of GCaMP6s, as well as the fact that multiple behavioral events co-occur within a trial, simply doing trial-type averaging is potentially problematic. For example, for the neuron show in Fig. 2B, it is arguable that the stimulus response itself (i.e. $< \sim 0.5$ s) is not different between trial types. Although it helps that only the 1-s response window was analyzed, it is still important to show that the modulation of visual responses by action is not due to (kinematic) aspects of the motor output other than the direction of turn.

To directly test for the effect of movement kinematics on ACC responses, we computed the mean instantaneous ball velocity on individual trials (calculated from movement onset until the ball reached the response threshold). We did a median split of left cue/ipsiversive and right cue/contraversive trials based on the velocity and quantified neuronal responses over a 1s window after stimulus onset (Supplementary Figure 3). This analysis showed that responses of ACC neurons are not modulated by ball velocity.

I believe a regression-type approach to quantify neural activity as a function of all behavioral events is warranted here.

We have carried out the reviewer's suggestion of a regression analysis to better isolate the effect of various behavioral events on neuronal responses. We built a regression model which describes the activity of each neuron as a combination of time-varying event kernels. We included left/right cues and ipsiversive/contraversive actions as event kernels in this model (Reviewer Figure 1). This analysis showed ipsiversive actions make a higher contribution than contraversive actions to ACC activity, as also seen in our other analyses (Fig. 2). However, we note that there are significant caveats in interpreting these results. In addition to the slow GCaMP6s dynamics pointed out by the reviewer, another issue is that most trials are correct. Hence, left cue/ipsiversive action and right cue/contraversive action are highly correlated, limiting the conclusions that can be drawn from this analysis.

Moreover, I would suggest plotting the average movement trajectories alongside with the activity plots.

We now include average ball position trajectories alongside the responses of example neurons shown in Figs 2 and 3.

Reviewer Figure 1. Regression analysis of ACC task responses.

A) We performed a regression analysis to model ACC task responses; right, left cues and contra, ipsi actions were used as time-varying event kernels for activity prediction. The cue kernel was supported from 0-1s relative to stimulus onset, whereas the choice kernel was supported -0.2s to 0.8s from choice. We used elastic net regularization (MATLAB function `lassoglm` with $\alpha = 0.5$ and $\lambda = 0.02$) and five-fold cross validation to prevent model overfitting. The time varying beta values for each action is shown. Only neurons with at least 2% variance explained were included in this analysis (353 neurons from 5 mice). B) To compute the relative contribution of each predictor to model performance, we first calculated the cross-validated variance explained by including all predictors in the model (R_{all}^2). We then dropped the beta values corresponding to one of the predictors, re-evaluated the model's predictions on test trials, and re-computed the cross-validated variance explained (R_{pred}^2). Relative contribution for each predictor was then computed as $1 - \frac{R_{pred}^2}{R_{all}^2}$. *** $p < 0.005$, Wilcoxon rank-sum test.

6) Moreover, in addition to quantifying the average magnitude of activity across trials (or even better, regression coefficients), the authors should also quantify the proportions of neurons that significantly respond to the different stimulus-action combinations.

To quantify the proportion of neurons responsive to different stimulus-action combinations, we compared responses over a 1s period after stimulus onset on trials with the same cue but with opposing actions using a Wilcoxon signed-rank test. This analysis identifies neurons whose cue-aligned responses are modulated by action direction. However, we note that in several sessions we have as few as 5 error trials, compromising the statistical power of this analysis. Hence, the proportions shown in Figs. 2H and 3E are only conservative estimates.

7) Fig 3. To strengthen the claim of the relevance of this ACC>SC projection, it would be important to show that action encoding is overrepresented in this population as compared to the overall population of ACC neurons (i.e. Fig 2). In Fig. 3E, are response magnitudes higher in this population as compared to the data in Fig. 2? What about proportions of neurons that significantly encode action? Is the action decoding accuracy from ACC>SC neurons (Fig. 3G) higher than the general ACC population?

To test if ACC-SC neurons contain more action information than the general ACC population, we performed decoding analysis on individual sessions (Figure 3G). We subsampled unlabeled neurons to match the number of retrogradely cells present in each session. We found that ACC-SC action decoding accuracy was higher than the unlabeled population in 7/8 sessions. We have added a description of these results and the associated methods in lines 175-179 and 800-807.

8) SC recordings, Fig. S4. This is important data to help interpret the results. It would be very interesting to further analyze the effects of ACC activation in the initial parts of the laser ON epoch (~1 s), matching the response window during the task. In particular, are there any differences in the prevalence, magnitude and time course of inhibited vs excited units in the SC during this initial period? This would get at the possibility that e.g. an early net inhibitory effect is followed by more prolonged excitation.

We have followed the reviewer's suggestion. In this experiment, recordings were made from intermediate and deep layers of the SC during presentation of visual sparse noise stimuli (also see lines 855-

Reviewer Figure 2. Time course of SC activity modulation with photostimulation of the ACC.

A) Laser modulation index of SC neurons that are excited (blue) and inhibited (red) during photoactivation of the ACC, plotted in 1s time bins from laser onset. Note that these are the same neurons as shown in Supplementary Figure 4C. B) Proportion of SC neurons that are excited and inhibited with photoactivation of the ACC in each time bin. Significant neurons were identified by comparing activity in laser and no-laser trials with a Wilcoxon test. C) Laser modulation index for neurons excited or inhibited during any of the time bins in B.

862 in Methods). While we did not observe retinotopically organized visual responses in these neurons, we reasoned that the potential non-specific increase in activity with visual stimulation would better enable us to detect inhibition of activity with ACC photostimulation. We classified individual SC neurons as excited or inhibited by comparing activity with and without laser photostimulation during visual stimulation (Supplementary Figure 6B, C). To investigate the temporal profile of ACC modulation of SC activity, we computed a laser modulation index (computed as $\frac{FR_{Laser} - FR_{nonlaser}}{FR_{Laser} + FR_{nonlaser}}$, where FR is the firing rate) in 1s bins following laser onset (Reviewer Figure 2A). This analysis shows that excited and inhibited neurons show relatively stable modulation across time. In addition, we quantified the proportion of neurons that are significantly excited and inhibited in 1s bins (Reviewer Figure 2B, C). There was a transient increase in the proportion of inhibited neurons in the 2nd time bin, which coincides with the onset of visual stimulation (which starts 0.5s after the laser). Furthermore, there were more positively than negatively modulated neurons in all time bins. Our results thus suggest that non-specific ACC activation both excites and inhibits subsets of SC neurons. As we discuss in the text (lines 332-348), these findings are consistent with multiple potential mechanisms for how the ACC might exert the opposite response bias to the SC in this task. However, our current results do not allow us to distinguish between the possibilities.

Minor issues:

1) Throughout the manuscript, the authors refer to ACC projections to motor layers of the SC as top-down input (e.g. 16, 54 etc). However, I would argue that “top-down input” is usually assumed to be comprised of feedback projections within a hierarchy, and is typically used to describe cortico-cortical connections. The author’s own data suggest, however, that the SC is a key node in the generation of the motor output itself, which many would argue would place it higher in a hierarchical scheme implementing sensorimotor transformations. To avoid confusion in readers, I would suggest replacing top-down input by simply input.

We have replaced ‘top-down input’ with ‘input’ when referring to ACC projections to the SC.

2) Rabies virus tracing in rostral and caudal ACC, line 85. It would help to add details here to help readers understand the strategy. Even after reading the methods, I am not entirely clear on how this experiment was setup. For instance, what mouse line was used? Do I understand correctly that this was an AAV with a CAG promoter injected in wild-type mice? If so, this would imply no cell-type specificity in the target regions, and if that is the case, why go for RV in the first place? To be clear, none of these concerns call into question the results, but please clarify the methodology.

We apologize for our unclear description of these experiments. These experiments did not use the EnvA-pseudotyped rabies virus which is used for cell-specific monosynaptic tracing and requires expression of the avian TVA receptor via AAV virus injection. We used the G-deleted rabies virus described by Wickersham et al., 2007 (PMID: 17179932), which acts as a retrograde tracer and is taken up by axons at the injection site. We used this virus because it leads to robust fluorophore expression throughout various neuronal compartments. However, the reviewer is correct that in principle these same experiments can also be performed using other retrograde tracing methods (such as CTB, retrobeads, or retro-AAV). We have modified the text and added a reference to the Wickersham et al. paper to better clarify the methodology for this experiment (lines 83-85): “We

performed retrograde tracing using modified rabies viruses⁵⁰ to identify sources of visual inputs to the ACC. Rabies viruses encoding GFP and tdTomato were injected into caudal and rostral ACC, respectively (Figure 1A).”

In our previous version, the methods for the rabies viruses mistakenly contained information for how an EnvA-pseudotyped virus would be generated (which we did not use; we used a B19G enveloped virus). Hence, we have removed the irrelevant portion of the description.

3) Laser effect panels in Figs 4 - 6. Please add more ticks to the y axis, it is presently difficult to visually estimate effect sizes.

We have added more tick marks in these figures.

4) Methods, line 763. For the sake of reproducibility, I would suggest spelling out the laser probability (or range thereof), rather than saying ~20%. Also, what does pseudorandom mean here, exactly? Is it balancing contra and ipsi trials? Please clarify.

We computed for each mouse the percent of trials that were inactivated and now specifically report these empirically determined values. We imposed the condition that two laser trials could not occur consecutively and hence referred to our procedure as pseudorandom. We have modified the text as follows to clarify our procedure (lines 636-638): “Across the mice, $30.4 \pm 5.1\%$ (mean \pm standard deviation; range 13.3% to 34.5%) of trials were inactivated. Trials were randomly selected for inactivation, except photostimulation could not occur on two consecutive trials.”

5) Methods, line 830. Most statistical tests in the manuscript used a permutation strategy, which is sound. Is there a reason why this one is a simple t-test?

We now report p-values for this analysis using a permutation test.

6) Methods, ROI selection, lines 858-865. More details are needed here. For example, what does it mean that maximum, mean and standard deviation projections were all used? Did an ROI have to present in all three to be counted? How is the neuropil area defined, e.g. is it a ring of n pixels around the ROI?

We now include additional details for this section of the Methods in lines 723 – 732: “ROIs were drawn manually over visually identified neuronal somas. We used reference frames generated by taking the pixel-wise maximum, mean, and standard deviation projection of all frames. These projections were compared with each other to visually identify neurons. In most cases, neurons were visible in all three projections; however, some neurons could be more readily identified in one of the three projections. For example, highly active neurons with a low baseline fluorescence could be difficult to identify in the mean projection but show up well in the standard deviation projection. Neurons had to be visually present in at least one of these projections to be included. We additionally verified that the visually selected neuronal soma had dynamic GCaMP6s signals. We defined the neuropil for each neuron by placing an ROI with the same shape as the somatic ROI over an adjacent area devoid of other neurons”.

Finally, was F0 a constant value throughout the session or was it computed over a sliding window? If the former, was there any other method apply to correct for baseline drifts?

The F0 value was a constant, but we applied the following procedure to correct for potential drifts in the baseline (lines 738-741): “To correct for drifts in baseline, we employed the method described in Pachitariu et al⁸⁰. We computed a moving baseline by filtering the data with a Gaussian of 1s width, then minimum filtering followed by maximum filtering with a window of 60s. The resulting baseline was subtracted from the DFF trace before z-scoring”.

7) Methods, line 955. What criteria were used in manual spike cluster curation?

We have added the following additional description to the methods (lines 842-854): “Single units sorted from the Mountain Sort (v3) algorithm were visualized with the MountainView software for manual curation. We first made a visual inspection to remove artifacts, which typically included large spikes observable in most channels (noise units) and symmetrical spike waveform of low amplitude with high firing frequency. Using spike auto-correlograms, we rejected all units with a large number of spikes happening in the refractory period of another spike (more than 0.25% of spikes occurring at < 1 ms after a spike event). By examining the amplitude histogram of each unit, we removed units with cropped amplitude distribution that could have resulted from thresholding. We finally merged single-units that were split into two units but had similar waveforms. To do so, we inspected their time series and merged two similar units that resulted from drift from the probe. In case of doubt whether two similar units are from the same neuron, we exclude them. For each resulting unit, we use only the portion of the recording where the baseline firing rate was constant.”

Reviewers' Comments:

Reviewer #1:

Remarks to the Author:

The authors have successfully addressed my questions. I recommend publication.

Reviewer #2:

Remarks to the Author:

The authors have addressed all of my concerns and the manuscript is much improved. I congratulate them on a very interesting paper.

Response to reviewers' comments

We thank the editor and the reviewers for their comments. Below, we provide responses to reviewer comments in blue.

Reviewer #1 (Remarks to the Author):

The authors have successfully addressed my questions. I recommend publication.

We thank the reviewer for their constructive feedback on our study.

Reviewer #2 (Remarks to the Author):

The authors have addressed all of my concerns and the manuscript is much improved. I congratulate them on a very interesting paper.

We are delighted that the reviewer finds our paper interesting and appreciate their comments which further strengthened the manuscript.